# Effects of Combined Application of Organic Fertilizer on the Growth and Yield of Pakchoi under Different Irrigation Water Types

**Shudong Lin, Chunhong Wang, Qingyuan Lei, Kai Wei, Quanjiu Wang *, Mingjiang Deng, Lijun Su** 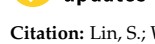**, Shiyao Liu and Xiaoxian Duan**

State Key Laboratory of Eco-Hydraulics in Northwest Arid Region, Xi'an University of Technology, Xi'an 710048, China; shudong_lin@163.com (S.L.)

* Correspondence: wquanjiu@163.com or aswe_lsdw@163.com

**Abstract:** The long-term utilization of inorganic fertilizers in pakchoi cultivation can result in increased nitrate levels, potentially posing health risks to human consumers. For this study, we investigated the efficacy of organic fertilizers as a promising alternative for enhancing soil structure, improving fertility, and increasing the yield of pakchoi. A two-year field trial was conducted from 2022 to 2023 to examine the effects of the combined application of organic fertilizer on the growth and yield of pakchoi. Three types of irrigation water, namely fresh water (F), brackish water (B), and magnetized–ionized brackish water (MIB), were used in combination with five different organic fertilizer rates (0, 20, 40, 60, and 80 kg/ha, denoted as 0, 1, 2, 3, and 4). The results revealed that treatments F2, F3, B2, B3, and MIB3 significantly improved the growth indexes of pakchoi. Notably, treatments F3, B3, and MIB3 resulted in an earlier onset of the fast growth period for leaf area index and fresh weight. During this period, we observed the highest cumulative growing degree days (ΔCGDD) values, which were 628.36 °C for plant height (MIB4), 475.01 °C for leaf area index (B3), 259.73 °C for fresh weight (B3), and 416.82 °C for dry matter accumulation (B3). The logistic model indicated an increase in eigenvalue at an organic fertilizer application rate of 60 kg/ha, while excessive fertilization had inhibitory effects. Under brackish water irrigation, both plant height and leaf area index demonstrated significant positive effects on yield, with plant height having a particularly noteworthy direct effect at a coefficient of 0.935. MIB water irrigation demonstrated superior advantages for promoting pakchoi growth, leading to significantly higher rates of fresh weight and dry matter accumulation compared to traditional brackish water irrigation. The maximum value of each growth index exerted a significant direct influence on its respective growth parameter, whereas ΔCGDD demonstrated a relatively smaller or potentially negative effect. Applying organic fertilizer appropriately can assist in the production of pakchoi and provide a scientific basis for increasing yield.

**Keywords:** pakchoi; organic fertilizer; growth dynamics; yield; environmental management

## 1. Introduction

Agricultural output is crucial for sustaining human survival, health, and nutrition [1]. Irrigation technology and fertilizer management are two technological or agrotechnical essential factors in agricultural production. The agricultural ecosystem, comprising water and fertilizer, forms the foundation of crop cultivation and animal husbandry [2,3]. As a result, water scarcity and low fertilizer utilization rates can significantly impact agricultural output, leading to food crises and jeopardizing human survival and development. However, in arid and semi-arid regions, limited water availability often restricts the use of fertilizers. Low fertilizer utilization rates can be attributed to water scarcity, making efficient nutrient management crucial for sustainable agriculture [4,5]. In addition, some

areas have inherently poor soil quality, which may require increased fertilizer application to enhance soil fertility and crop productivity. Conversely, regions with naturally fertile soils might face issues related to excessive fertilizer use [6]. Despite the world's abundant water resources, only a small portion consists of freshwater. This discrepancy between water supply and demand poses a severe challenge [7,8].

With the global economy advancing and human living standards improving, modernization, science, and technology are gradually transforming agricultural production towards intelligence. However, we cannot overlook the challenges that persist in agricultural production. The lack of scientific irrigation and fertilization management is causing secondary salinization of soil in many areas [9,10]. Although soil salinization is a natural occurrence, it has been exacerbated by long-term human activities. Currently, globally recorded secondary salinized land covers about 77 million ha, of which 58% is in agricultural irrigated areas [11]. The use of traditional methods of irrigation leads to water waste and soil salinization, whereas traditional fertilizer use decreases land fertility and causes the accumulation of chemical residues, resulting in environmental pollution [2,10]. Therefore, improving water and fertilizer utilization efficiency and reducing environmental impacts are urgent issues that modern agriculture must address.

Contemporary extensive and in-depth research conducted by scholars in the field ahs aimed to enhance the efficiency of agricultural water and fertilizer resources and effectively mitigate soil salinization. Researchers have discovered irrigation water activation technology, which can facilitate a water treatment approach suitable for agricultural irrigation that involves magnetized treatment and ionized treatment [12–14]. On one hand, magnetized water irrigation combined with fertilizer application has shown a better growth promotion effect on cotton [15,16], tomato [17], corn [10], rice [18], and other crops. It can enhance seed emergence rates, reduce soil salt content, improve the salt tolerance and stress resistance of crops, increase crop productivity, and enhance the quality of crops and fruits [19,20]. Similarly, the application of ionized technology in irrigation with brackish water has yielded positive outcomes in promoting growth and increasing crop production [14,21]. Ionized water irrigation can improve the growing environment of crops, increase the growth rate and yield of winter wheat, and have a certain buffer effect on soil salt stress [22]. Additionally, ionized irrigation combined with N, P, K can reduce salt stress and improve soil water holding capacity [23]. Researchers have also investigated the combined treatment of magnetized and ionized water. The results of one study indicated that magnetized–ionized water can enhance the leaching efficiency and water usage of soil salt [24]. These findings suggest that activated water treatment technology can modify the physical and chemical properties of both freshwater and brackish water, thereby enhancing the efficiency of irrigation water utilization and its potential application in agricultural production.

On the other hand, an inorganic fertilizer containing abundant nutrients can enhance crop yield and improve quality [25,26]. However, the excessive and prolonged use of traditional chemical fertilizers can result in decreased fertilizer efficiency, the degradation of soil physical and chemical properties, elevated agricultural production costs, and severe environmental pollution [27,28]. The excessive accumulation of nitrate nitrogen in soil can result in increased nitrate content in vegetable products, leading to a decline in quality and posing a risk to human health [29]. The results of one study in there literature indicated a positive correlation between fertilizer application and the nitrate content of vegetables, but excessive fertilization did not promote yield increase; instead, it decreased yield and affected the nutritional quality of vegetables. Several alternatives have been recommended, including the use of organic fertilizers. Organic fertilizer is a natural fertilizer produced through composting, decay, and other similar methods. Organic fertilizers supply nutrients for crop growth and are abundant in organic matter; they also supply essential micronutrients that are crucial for healthy crop development and typically absent in inorganic fertilizers [30,31]. In recent years, numerous studies have focused on the combined application of organic fertilizer in crop growth, yield, soil fertility, and nutrient utilization [28,32]. Scholars from home and abroad have conducted extensive research on wheat [33], corn [34],

rice [35], and other crops under organic fertilizer application conditions. These studies have shown that the appropriate application of organic fertilizer can enhance crop growth, improve yield composition, increase crop yield, augment soil nutrient content, and enhance fertilizer utilization rates. Moreover, organic fertilizers contain a high concentration of organic matter and microbial flora, which can enhance soil fertility, improve soil structure, and stimulate plant growth and development [36].

Pakchoi is a popular leafy vegetable renowned for its short growth cycle, high yield, delightful taste, and nutritional richness [37]. While it is a great source of various vitamins, minerals, and cellulose, it can also pose a potential health risk due to its high nitrate content [29]. Nitrate accumulation in pakchoi can occur due to several reasons, with fertilization being a primary factor [25,38]. This is primarily due to the fact that the uptake of nitrate nitrogen exceeds the assimilation capacity of the pakchoi [39]. Therefore, it is essential to reduce nitrate levels and improve quality while sustaining high yield [40]. Domestic and international researchers have conducted numerous studies on rational fertilization, balanced fertilization, and the application of organic fertilizers to achieve this objective, yielding varying degrees of success [25,41,42]. However, few studies have investigated the effects of organic fertilizer on pakchoi growth under diverse irrigation water types. Hence, conducting comprehensive research on this topic is imperative. The aims of this paper are (i) to explore the influence of organic fertilizer on the growth index of pakchoi and the mechanism underlying its effects on yield difference, distribution in plant height, leaf area index, and fresh weight and dry matter accumulation, along with the relationships among these parameters under various irrigation water types; and (ii) to guide the application of organic fertilizer in the production of pakchoi, providing a scientific basis and technical support for its cultivation and contributing to the sustainable development of agricultural production.

## 2. Materials and Methods

### 2.1. Experimental Site and Description

The study was conducted in the Bazhou Irrigation Experiment Station in Korla City, Xinjiang, China (41°45′20.24″ N, 86°8′51.16″ E, 901 m above sea level). The region falls within a typical warm temperate zone characterized by a continental arid climate. The research was carried out between 2022 and 2023 (May to June), taking into account the specific climatic conditions of the area. The study site experiences a frost-free period of 175–200 days and benefits from 2173 to 3059 h of sunshine annually. To provide a comprehensive understanding of the soil, Table 1 presents the initial physical and chemical properties of the soil prior to the commencement of the experiment. We collected daily average temperature, maximum temperature, minimum temperature, and precipitation values from an automatic meteorological station (Davis Instruments, Hayward, USA, Davis VantagePro-6152) located in the field (Figure 1).

**Table 1.** Initial physical and chemical properties of soil.

| Depth (cm) | Bulk Density (g·cm$^{-3}$) | Mechanical Composition (%) | | | pH | Alkali-Hydrolyzed Nitrogen (mg·kg$^{-1}$) | Available Phosphorus (mg·kg$^{-1}$) | Available Potassium (mg·kg$^{-1}$) | Organic Matter (g·kg$^{-1}$) |
|---|---|---|---|---|---|---|---|---|---|
| | | Clay | Silt | Sand | | | | | |
| 0–20 | 1.45 | 5.3 | 52.5 | 42.2 | 8.82 | 24.34 | 41.37 | 240 | 8.10 |
| 20–40 | 1.37 | 3.3 | 32.9 | 63.8 | 8.15 | 13.83 | 8.79 | 144 | 5.10 |

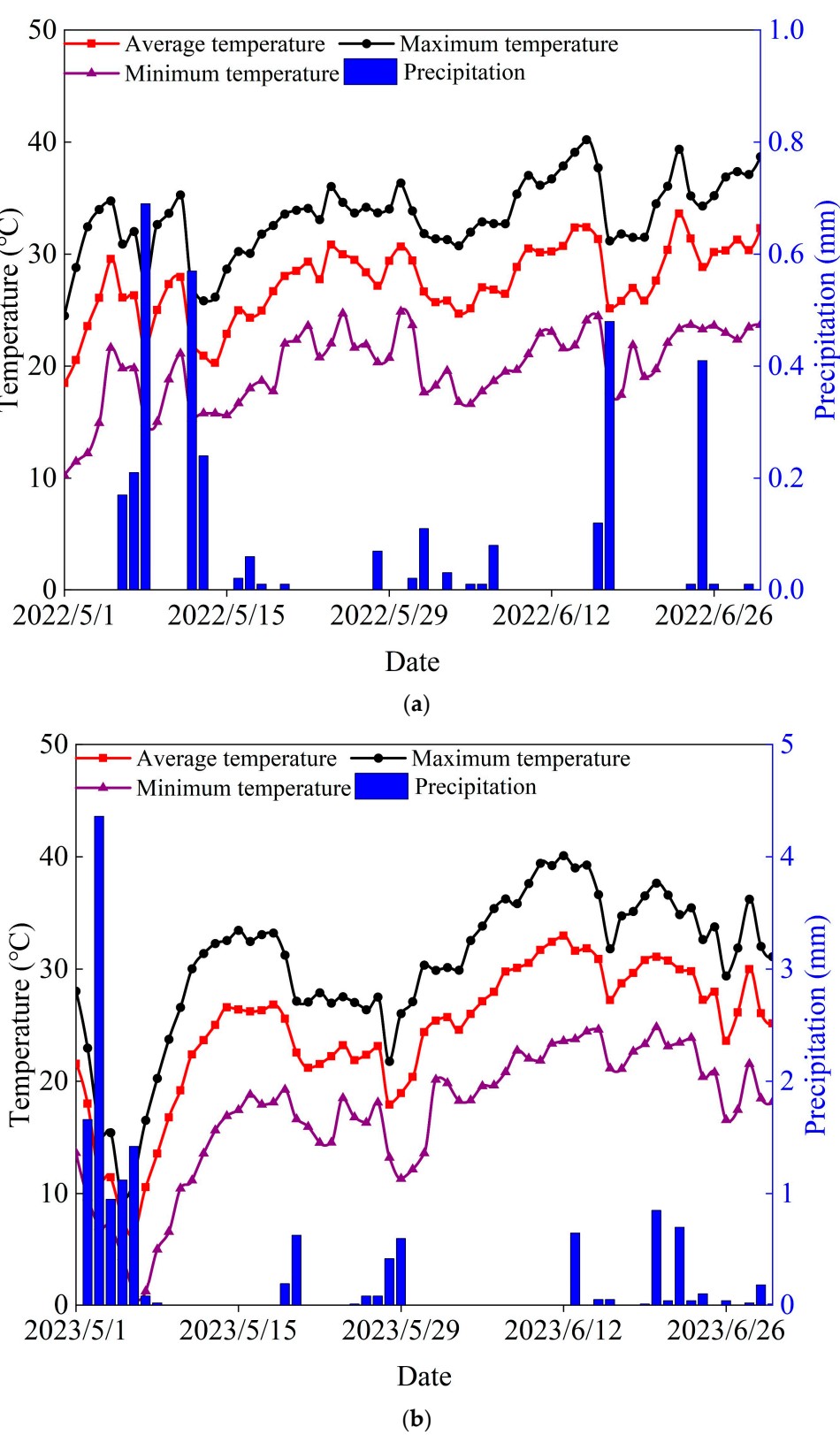

**Figure 1.** Temperature and rainfall during the whole growth period of pakchoi. (**a**) Temperature and rainfall in 2022. (**b**) Temperature and rainfall in 2023.

### 2.2. Experimental Design and Field Management

Pakchoi without heading formation, known for its fast growth, short cycle, and rich nutritional content in all four seasons, was selected for the study. For the experiment, the

traditional irrigation amount was about 450 m$^3$/ha (based on the annual water requirement for pakchoi in the area). A water-soluble biological organic fertilizer (Shidiji biological organic fertilizer, Chengdu Huahong Biotechnology Co., Ltd., Chengdu, China) containing organic matter ($\geq$40%), N + P$_2$O$_5$ + K$_2$O ($\geq$5%), amino acid ($\geq$10%), and fulvic acid ($\geq$5%) was used. Effective live bacterial count: 525 million/g; Bacillus amyloliquefaciens: 420 million/g; Bacillus subtilis: 100 million/g; Bacillus mucilaginosus: 4.7 million/g; pH: 5.5; Total lead: 5.2 mg/kg; Total chromium: 2.7 mg/kg. The organic fertilizer was applied with water during eight irrigations throughout the growth period. In this study, we applied the fertilizer at regular intervals throughout the growth period of the pakchoi plants. Specifically, we applied the fertilizer once every five days during the growing season. The experiments used a flood irrigation system and involved five levels of organic fertilizer application rates (0, 20, 40, 60, and 80 kg/ha) and three types of water (fresh water, F; brackish water, B; and magnetized–ionized brackish water, MIB). This resulted in the treatments being labeled as follows: F0, F1, F2, F3, and F4 (0, 20, 40, 60, and 80 kg/ha); B0, B1, B2, B3, and B4 (0, 20, 40, 60, and 80 kg/ha); and MIB0, MIB1, MIB2, MIB3, and MIB4 (0, 20, 40, 60, and 80 kg/ha).

The seeds were sown on 7 May and 5 May in 2022 and 2023, respectively. The harvest dates in 2022 and 2023 were 30 June and 28 June, respectively. In this study, we adopted the standard sowing method for pakchoi cultivation. The seeds were directly sown in the field where the main experiment was conducted. At the beginning of the planting season, the pakchoi seeds were directly sown into the prepared soil beds. Following standard agronomic practices, the seeds were sown at a depth of approximately 3–4 cm in the soil. Prior to sowing, 300 kg/ha of compound fertilizer (tert.-ammonium, N-P$_2$O$_5$-K$_2$O, 19%–19%–19%) was applied in accordance with local practices, and no additional fertilizer was used during the experiment. Each experimental plot was enclosed by a high-quality, thickened, waterproof sheet, 2 m in depth. The fertilizer treatment plot was fixed for 2 years and repeated 3 times in a randomized block design. The experimental plot area was 3 m$^2$; the planting distance for pakchoi was approximately 20 cm between plants, with rows spaced approximately 40 cm apart, with a planting density of 2.5 × 10$^5$ plants/ha. Weeding, pest and disease control, and other field management were carried out in accordance with local practices.

*2.3. Measurements and Calculations*

2.3.1. Pakchoi Growth Index

During the entire growth period, four plants were selected randomly from each plot. Data collection commenced twelve days after seeding, with subsequent measurements taken at ten-day intervals throughout the experiment. The plant height (PH) was measured using a tape measure, and the length and width of each leaf were measured in accordance with the method described by Kumar [43] to determine the leaf area. The green leaf area of each plant was calculated using the following formula [44]:

$$LA = \sum_{i}^{n} 0.75 \times A_i \times B_i \tag{1}$$

where *LA* represents the leaf area of per plant of pakchoi (cm$^2$), $A_i$ (cm) and $B_i$ (cm) are the length and width of a leaf of pakchoi, *n* is the number of leaves per plant of pakchoi, and 0.75 is the correction factor for pakchoi [44]. The *LAI* was then obtained as follows [3,45]:

$$LAI = \frac{LA_t}{S_l} \tag{2}$$

where $LA_t$ is the total area of the pakchoi leaves (cm$^2$); $S_l$ is the occupied land area (cm$^2$).

After the whole growth period, four plants were randomly selected from each plot. After measuring the fresh weight with electronic balance, the leaves and stem of the plants were placed into an oven at 105 °C for 30 min and then dried at 75 °C to a constant weight;

they were subsequently weighed. The fresh weight and dry matter accumulation was determined as follows:

$$FW = FW_p \times P \tag{3}$$

$$DM = DM_p \times P \tag{4}$$

where $FW$, $DM$ is the fresh weight and dry matter accumulation (kg/ha); $FW_p$, $DM_p$ is the average fresh weight and dry matter accumulation of per plants (kg/plant); P is plant density (plants/ha). The yield of pakchoi at the maturity stage was determined according to its fresh weight. To calculate the yield using Equation (3), all of the pakchoi plants in each plot were harvested and weighed.

### 2.3.2. Pakchoi Growth Model Based on Cumulative Growing Degree Days

Each crop had a specific range of biological upper and lower temperature limits, beyond which its growth will be hindered. Previous research has demonstrated that the upper and lower temperature limits for pakchoi are 35 °C and 7 °C, respectively [46–48]. The number of growing degree days (GDD) required by the same crop to complete its growth cycle is relatively consistent within a given region. The growth period of a crop can be expressed using GDD, assuming suitable temperature and other environmental conditions [49]. Local growth models have been established in previous studies using GDD. GDD represents the difference between the daily average temperature and the minimum temperature required for crop activities. In this study, daily GDD data were converted to cumulative GDD ($CGDD$) using the following equation:

$$CGDD = \sum (T_{avg} - T_{base}) \tag{5}$$

where $CGDD$ is the cumulative growing degree days (°C), $T_{avg}$ is the mean daily temperature, and $T_{base}$ is the minimum daily temperature required for crop activity. McMaster and Wilhelm proposed a method for calculating $T_{avg}$ [49]:

$$T_{avg} = \begin{cases} \frac{(T_x + T_n)}{2} & if\ T_{base} < T_{avg} < T_{upper} \\ T_{base} & if\ T_{avg} \leq T_{base} \\ T_{upper} & if\ T_{avg} \geq T_{upper} \end{cases} \tag{6}$$

where $T_{upper}$ is the highest temperature required for crop activities (°C), $T_x$ is the highest daily temperature (°C), and $T_n$ is the lowest daily temperature (°C).

The logistic model describing the plant height ($PH$), leaf area index ($LAI$), fresh weight ($FW$) and dry matter accumulation ($DM$) of the crop with $CGDD$ as independent variable is established. A logistic model between the $CGDD$ and $PH$, $LAI$, and $FW$ and $DM$ can be expressed as:

$$Y = \frac{Y_{max}}{1 + e^{a + b \times CGDD}} \tag{7}$$

where $Y$ represents the plant height (cm), leaf area index (cm$^2$/cm$^2$), and fresh weight (kg/ha) and dry matter accumulation (kg/ha), respectively. $Y_{max}$ is the maximum value of $Y$.

The growth rate equation can be obtained by taking the first order derivative of Equation (7):

$$V = \frac{dY}{dCGDD} = -\frac{Y_{max} b e^{a + b \times CGDD}}{\left(1 + e^{a + b \times CGDD}\right)^2} \tag{8}$$

To obtain the cumulative growing degree days at the maximum growth rate ($CGDD_0$), we need to take the second-order derivative of Equation (7) and set it equal to 0:

$$CGDD_0 = -\frac{a}{b} \tag{9}$$

Therefore, we can obtain the maximum growth rate ($V_{max}$) from Equation (8) by substituting it into Equation (9). By substituting Equation (9) into Equation (7), the corresponding value of the growth index at the maximum growth rate ($Y_{Vmax}$) can be obtained. The growth rate equation has two inflection points which can be used to divide the pre, mid, and late stages of the growth process. The gradual growth period is defined as the period before $CGDD_1$, while the fast growth period begins at $CGDD_1$ and ends at $CGDD_2$. The slow growth period follows $CGDD_2$. Therefore, the eigenvalues of $CGDD_1$, $CGDD_2$, $\Delta CGDD$, and $V_{avg}$ represent the initiation cumulative growing degree days, termination cumulative growing degree days, duration cumulative growing degree days, and average growth rates during the fast growth period of pakchoi, respectively.

$$CGDD_1 = \frac{\ln(2 + \sqrt{3}) - a}{b} \tag{10}$$

$$CGDD_2 = \frac{\ln(2 - \sqrt{3}) - a}{b} \tag{11}$$

$$\Delta CGDD = CGDD_2 - CGDD_1 \tag{12}$$

$$V_{avg} = \frac{\int_{CGDD_1}^{CGDD_2} V dCGDD}{\Delta CGDD} = \frac{Y_{\max}\left(e^{a+b \times CGDD_1} - e^{a+b \times CGDD_2}\right)}{\Delta CGDD \left(1 + e^{a+b \times CGDD_2}\right)\left(1 + e^{a+b \times CGDD_1}\right)} \tag{13}$$

### 2.4. Data Processing Method and Error Analysis

All data processing and plotting were conducted using Microsoft Office Excel (2016, Microsoft Corporation, Seattle, USA) and Origin, respectively. MATLAB (2014, MathWorks Inc., Natick, MA, USA) programming was utilized to establish and solve the model. Correlations were tested using the $R^2$; the accuracy was evaluated through using root mean square error ($RMSE$), and normalized root mean square error ($nRMSE$).

$$R^2 = \frac{\left[\sum (x_i - \overline{x})(y_i - \overline{y})\right]^2}{\sum (x_i - \overline{x})^2 \sum (y_i - \overline{y})^2} \tag{14}$$

$$RMSE = \sqrt{\frac{\sum (m_{vi} - c_{vi})^2}{n}} \tag{15}$$

$$nRMSE = \frac{RMSE}{\overline{m_{vi}}} \times 100\% \tag{16}$$

where $x_i$ is the independent variable, $y_i$ is the dependent variable, and $\overline{x}$ and $\overline{y}$ represent the average value of $x_i$ and $y_i$; $m_{vi}$ is the measured value, and $c_{vi}$ is the calculated value.

## 3. Results

### 3.1. Evaluation of Growth Index Logistic Model Fitting

Figures 2 and 3 display the plant height (Figure 2a–f) and leaf area index (Figure 2g–l) and fresh weight (Figure 3a–f) and dry matter accumulation (Figure 3g–l) characteristics in response to different organic fertilizer application rates and irrigation water types across CGDD. The growth index of pakchoi exhibited a typical pattern of accumulation characterized by a "slow-quick-slow" increase with increasing CGDD, highlighting the dynamic nature of pakchoi growth under the influence of organic fertilizer and irrigation water types.

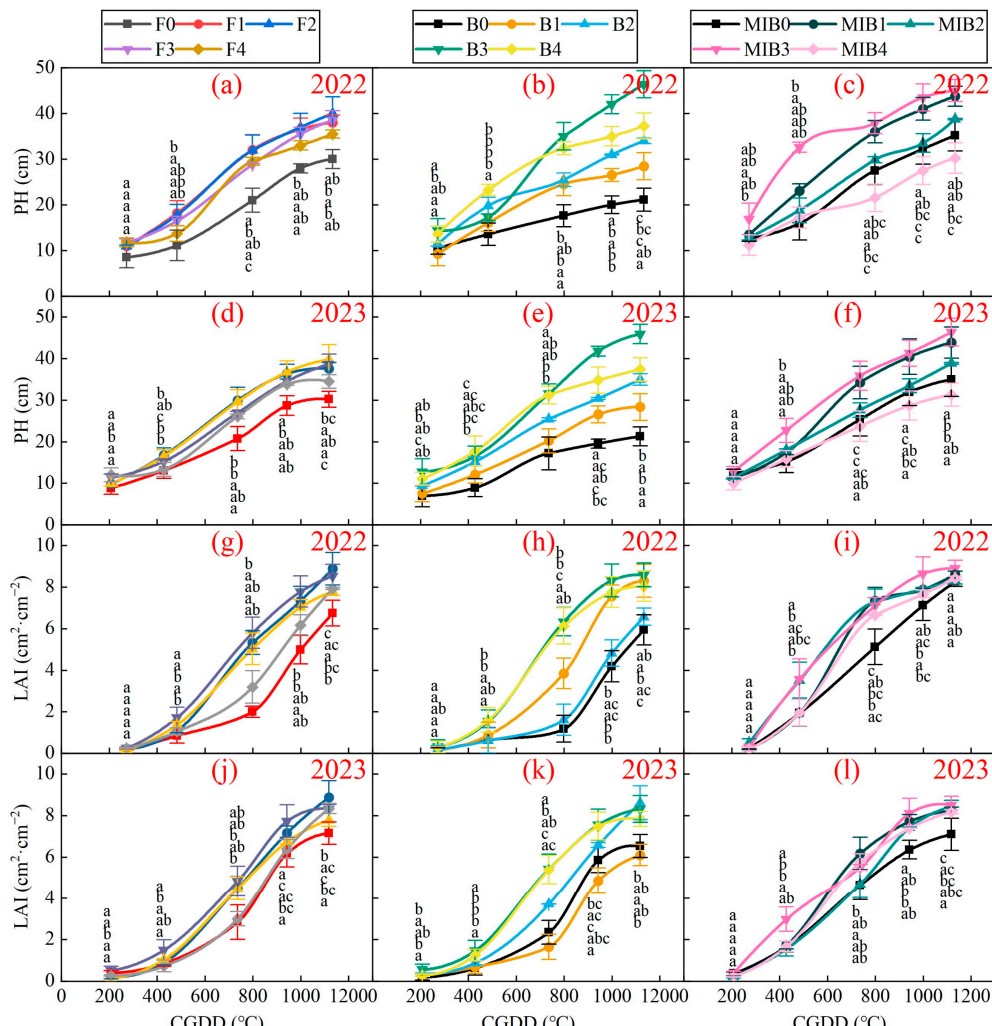

**Figure 2.** Plant height and leaf area index compared with cumulative growing degree days in 2022 and 2023. Different letters indicate significant differences among treatments at *p* < 0.05.

The application of F2 and F3 significantly increased the growth index of pakchoi when irrigated with fresh water, while B2 and B3 significantly improved the growth index under brackish water irrigation. MIB3 showed a significant increase in the growth index when irrigated with magnetized–ionized brackish water. B4 significantly increased the plant height of pakchoi during the early growth stage, whereas B3 showed significant improvement in plant height during the late growth stage. However, MIB4 had an inhibitory effect on pakchoi plant height throughout the growth period under magnetized–ionized brackish water irrigation. Furthermore, when the amount of organic fertilizer applied is the same, magnetized–ionized brackish water irrigation is more conducive to the growth of pakchoi. In summary, to optimize the growth of pakchoi, it is crucial to carefully select both the type of organic fertilizer and the irrigation water. Our study indicates that an organic fertilizer application rate of 60 kg/ha is particularly beneficial for enhancing pakchoi growth.

The logistic model was used to fit the dynamic accumulation process of growth indexes. The equations and test indicators are presented in Tables 2 and 3, demonstrating an $R^2$ range of 0.928–0.999 and nRMSE of 2.004–14.965%. These results indicate that the logistic model is effective in simulating the dynamic accumulation of growth indexes. Additionally, the various biologically significant characteristic parameters derived from the logistic model can be used to analyze the effect of different organic fertilizer gradients on the dynamic accumulation of growth indicators of pakchoi under various irrigation water types.

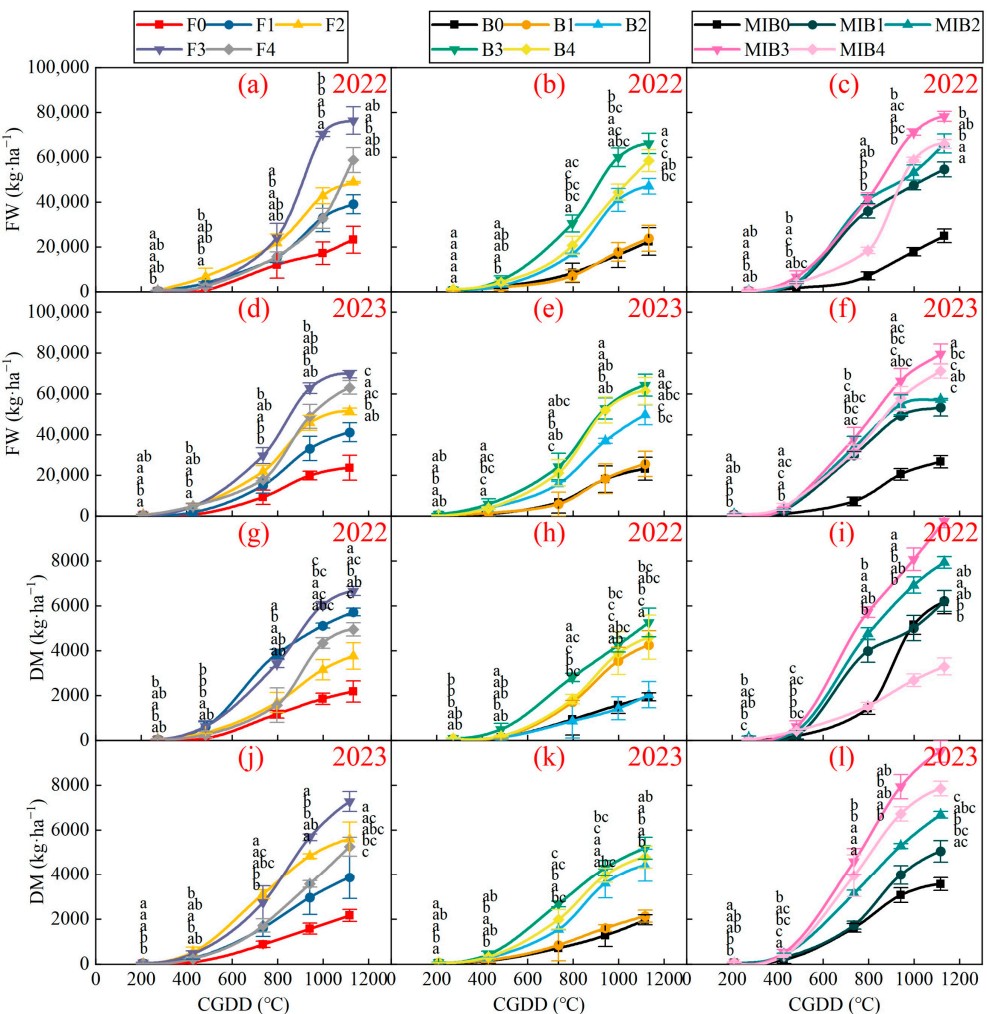

**Figure 3.** Fresh weight and dry matter accumulation compared with cumulative growing degree days in 2022 and 2023. Different letters indicate significant differences among treatments at $p < 0.05$.

**Table 2.** Logistic model fitting results for the dynamic change process of plant height and leaf area index in 2022 and 2023.

| Index | Treatment | 2022 | | | | 2023 | | | |
|---|---|---|---|---|---|---|---|---|---|
| | | Logistic Model | $R^2$ | RMSE | nRMSE/% | Logistic Model | $R^2$ | RMSE | nRMSE/% |
| PH | F0 | $Y = \frac{30.85}{1+e^{2.932-0.00545 \times CGDD}}$ | 0.958 | 1.888 | 9.57 | $Y = \frac{30.75}{1+e^{2.555-0.00542 \times CGDD}}$ | 0.954 | 2.028 | 9.99 |
| | F1 | $Y = \frac{38.81}{1+e^{2.587-0.0053 \times CGDD}}$ | 0.993 | 1.152 | 4.25 | $Y = \frac{38.44}{1+e^{2.457-0.0053 \times CGDD}}$ | 0.990 | 1.443 | 5.54 |
| | F2 | $Y = \frac{40.39}{1+e^{2.669-0.0052 \times CGDD}}$ | 0.989 | 1.544 | 5.61 | $Y = \frac{40.18}{1+e^{2.545-0.00518 \times CGDD}}$ | 0.985 | 1.804 | 6.81 |
| | F3 | $Y = \frac{38.67}{1+e^{2.619-0.00503 \times CGDD}}$ | 0.969 | 2.405 | 9.18 | $Y = \frac{38.75}{1+e^{2.418-0.00484 \times CGDD}}$ | 0.959 | 2.467 | 9.74 |
| | F4 | $Y = \frac{36.52}{1+e^{2.629-0.00502 \times CGDD}}$ | 0.957 | 2.276 | 9.22 | $Y = \frac{35.42}{1+e^{2.413-0.00508 \times CGDD}}$ | 0.928 | 2.347 | 9.81 |
| | B0 | $Y = \frac{21.73}{1+e^{2.18-0.00484 \times CGDD}}$ | 0.984 | 0.650 | 3.93 | $Y = \frac{22.35}{1+e^{2.41-0.00504 \times CGDD}}$ | 0.977 | 1.124 | 7.62 |
| | B1 | $Y = \frac{29.23}{1+e^{2.002-0.00454 \times CGDD}}$ | 0.998 | 0.420 | 2.00 | $Y = \frac{29.06}{1+e^{2.027-0.0043 \times CGDD}}$ | 0.977 | 1.586 | 8.38 |
| | B2 | $Y = \frac{33.65}{1+e^{1.908-0.00443 \times CGDD}}$ | 0.966 | 1.934 | 7.95 | $Y = \frac{34.89}{1+e^{2.258-0.0047 \times CGDD}}$ | 0.980 | 1.745 | 7.57 |
| | B3 | $Y = \frac{46.57}{1+e^{2.023-0.00519 \times CGDD}}$ | 0.952 | 3.053 | 9.85 | $Y = \frac{46.28}{1+e^{2.731-0.00519 \times CGDD}}$ | 0.953 | 2.895 | 9.76 |
| | B4 | $Y = \frac{37.85}{1+e^{1.791-0.00456 \times CGDD}}$ | 0.997 | 0.617 | 2.18 | $Y = \frac{38.15}{1+e^{2.112-0.00488 \times CGDD}}$ | 0.992 | 1.170 | 4.43 |
| | MIB0 | $Y = \frac{35.42}{1+e^{2.449-0.00505 \times CGDD}}$ | 0.959 | 2.314 | 9.37 | $Y = \frac{35.14}{1+e^{2.566-0.00526 \times CGDD}}$ | 0.942 | 2.362 | 9.85 |
| | MIB1 | $Y = \frac{43.92}{1+e^{2.319-0.00507 \times CGDD}}$ | 0.993 | 1.214 | 3.86 | $Y = \frac{44.46}{1+e^{2.031-0.00452 \times CGDD}}$ | 0.984 | 2.068 | 7.03 |
| | MIB2 | $Y = \frac{38.18}{1+e^{2.154-0.00456 \times CGDD}}$ | 0.970 | 2.124 | 7.94 | $Y = \frac{38.15}{1+e^{2.09-0.00461 \times CGDD}}$ | 0.967 | 2.351 | 9.09 |
| | MIB3 | $Y = \frac{45.3}{1+e^{1.612-0.00465 \times CGDD}}$ | 0.962 | 2.561 | 7.28 | $Y = \frac{45.87}{1+e^{2.119-0.0049 \times CGDD}}$ | 0.987 | 1.790 | 5.63 |
| | MIB4 | $Y = \frac{29.84}{1+e^{1.79-0.00419 \times CGDD}}$ | 0.940 | 1.878 | 8.72 | $Y = \frac{31.52}{1+e^{1.888-0.00447 \times CGDD}}$ | 0.983 | 1.329 | 6.07 |

**Table 2.** *Cont.*

| Index | Treatment | 2022 Logistic Model | $R^2$ | RMSE | nRMSE/% | 2023 Logistic Model | $R^2$ | RMSE | nRMSE/% |
|---|---|---|---|---|---|---|---|---|---|
| LAI | F0 | $Y = \dfrac{7.62}{1+e^{6.153-0.00688\times CGDD}}$ | 0.992 | 0.286 | 9.72 | $Y = \dfrac{7.9}{1+e^{5.317-0.00682\times CGDD}}$ | 0.989 | 0.345 | 9.92 |
| | F1 | $Y = \dfrac{9.37}{1+e^{5.232-0.00679\times CGDD}}$ | 0.996 | 0.295 | 6.45 | $Y = \dfrac{9.61}{1+e^{5.058-0.00662\times CGDD}}$ | 0.999 | 0.156 | 3.60 |
| | F2 | $Y = \dfrac{8.23}{1+e^{4.981-0.0068\times CGDD}}$ | 0.999 | 0.125 | 2.94 | $Y = \dfrac{8.24}{1+e^{4.851-0.00679\times CGDD}}$ | 0.999 | 0.094 | 2.34 |
| | F3 | $Y = \dfrac{9.03}{1+e^{4.676-0.00657\times CGDD}}$ | 0.999 | 0.148 | 3.09 | $Y = \dfrac{9.4}{1+e^{4.065-0.00568\times CGDD}}$ | 0.996 | 0.256 | 5.61 |
| | F4 | $Y = \dfrac{8.8}{1+e^{5.685-0.00625\times CGDD}}$ | 0.996 | 0.244 | 6.58 | $Y = \dfrac{9.14}{1+e^{5.388-0.00622\times CGDD}}$ | 0.999 | 0.133 | 3.56 |
| | B0 | $Y = \dfrac{7.14}{1+e^{6.671-0.00721\times CGDD}}$ | 0.986 | 0.235 | 9.78 | $Y = \dfrac{7.91}{1+e^{5.548-0.0065\times CGDD}}$ | 0.983 | 0.305 | 9.86 |
| | B1 | $Y = \dfrac{9.51}{1+e^{6.023-0.00717\times CGDD}}$ | 0.996 | 0.276 | 6.66 | $Y = \dfrac{8.57}{1+e^{5.292-0.00635\times CGDD}}$ | 0.979 | 0.262 | 9.70 |
| | B2 | $Y = \dfrac{7.21}{1+e^{6.412-0.0062\times CGDD}}$ | 0.992 | 0.265 | 9.63 | $Y = \dfrac{9.55}{1+e^{4.94-0.00593\times CGDD}}$ | 0.999 | 0.095 | 2.36 |
| | B3 | $Y = \dfrac{9.29}{1+e^{4.618-0.00661\times CGDD}}$ | 0.996 | 0.273 | 5.44 | $Y = \dfrac{9.23}{1+e^{3.814-0.00555\times CGDD}}$ | 0.997 | 0.237 | 5.09 |
| | B4 | $Y = \dfrac{8.66}{1+e^{4.454-0.0065\times CGDD}}$ | 0.995 | 0.286 | 6.02 | $Y = \dfrac{8.53}{1+e^{4.199-0.00631\times CGDD}}$ | 0.995 | 0.298 | 6.70 |
| | MIB0 | $Y = \dfrac{8.69}{1+e^{4.558-0.00654\times CGDD}}$ | 0.993 | 0.330 | 7.28 | $Y = \dfrac{7.73}{1+e^{4.163-0.00648\times CGDD}}$ | 0.999 | 0.133 | 3.32 |
| | MIB1 | $Y = \dfrac{9.01}{1+e^{4.293-0.00667\times CGDD}}$ | 0.984 | 0.509 | 9.79 | $Y = \dfrac{8.79}{1+e^{3.879-0.00619\times CGDD}}$ | 0.994 | 0.330 | 6.85 |
| | MIB2 | $Y = \dfrac{8.85}{1+e^{3.229-0.00558\times CGDD}}$ | 0.977 | 0.531 | 9.66 | $Y = \dfrac{9.08}{1+e^{4.009-0.0063\times CGDD}}$ | 0.998 | 0.207 | 4.65 |
| | MIB3 | $Y = \dfrac{9.23}{1+e^{3.71-0.00626\times CGDD}}$ | 0.984 | 0.545 | 9.57 | $Y = \dfrac{9.02}{1+e^{3.477-0.00566\times CGDD}}$ | 0.980 | 0.506 | 9.95 |
| | MIB4 | $Y = \dfrac{8.71}{1+e^{4.401-0.00671\times CGDD}}$ | 0.992 | 0.369 | 7.42 | $Y = \dfrac{8.61}{1+e^{4.013-0.00624\times CGDD}}$ | 0.997 | 0.234 | 5.05 |

**Table 3.** Logistic model fitting results for the dynamic change process of fresh weight and dry matter accumulation in 2022 and 2023.

| Index | Treatment | 2022 Logistic Model | $R^2$ | RMSE | nRMSE/% | 2023 Logistic Model | $R^2$ | RMSE | nRMSE/% |
|---|---|---|---|---|---|---|---|---|---|
| FW | F0 | $Y = \dfrac{20930}{1+e^{11.84-0.0148\times CGDD}}$ | 0.965 | 2020.0 | 14.78 | $Y = \dfrac{23020}{1+e^{10.551-0.01315\times CGDD}}$ | 0.995 | 906.4 | 8.38 |
| | F1 | $Y = \dfrac{37940}{1+e^{10.91-0.01393\times CGDD}}$ | 0.984 | 2083.0 | 11.36 | $Y = \dfrac{40360}{1+e^{10.49-0.01363\times CGDD}}$ | 0.994 | 1680.0 | 9.18 |
| | F2 | $Y = \dfrac{47610}{1+e^{11.73-0.01447\times CGDD}}$ | 0.973 | 3515.0 | 14.56 | $Y = \dfrac{52140}{1+e^{8.542-0.01118\times CGDD}}$ | 0.994 | 2066.0 | 8.31 |
| | F3 | $Y = \dfrac{76620}{1+e^{13.82-0.01634\times CGDD}}$ | 0.999 | 1481.0 | 4.26 | $Y = \dfrac{71200}{1+e^{8.417-0.01101\times CGDD}}$ | 0.998 | 1816.0 | 5.42 |
| | F4 | $Y = \dfrac{63880}{1+e^{14.02-0.0143\times CGDD}}$ | 0.937 | 3214.0 | 14.58 | $Y = \dfrac{60540}{1+e^{11.73-0.01454\times CGDD}}$ | 0.982 | 4037.0 | 14.92 |
| | B0 | $Y = \dfrac{21260}{1+e^{10.997-0.01294\times CGDD}}$ | 0.961 | 1424.0 | 14.18 | $Y = \dfrac{22850}{1+e^{11.15-0.01309\times CGDD}}$ | 0.992 | 1076.0 | 10.86 |
| | B1 | $Y = \dfrac{24910}{1+e^{10.63-0.01261\times CGDD}}$ | 0.988 | 1278.0 | 12.50 | $Y = \dfrac{25600}{1+e^{10.68-0.0131\times CGDD}}$ | 0.988 | 1404.0 | 13.55 |
| | B2 | $Y = \dfrac{46690}{1+e^{11.97-0.01426\times CGDD}}$ | 0.995 | 1758.0 | 8.10 | $Y = \dfrac{46860}{1+e^{11.11-0.01092\times CGDD}}$ | 0.973 | 3043.0 | 14.22 |
| | B3 | $Y = \dfrac{67140}{1+e^{9.651-0.01187\times CGDD}}$ | 0.996 | 2328.0 | 7.15 | $Y = \dfrac{65580}{1+e^{8.007-0.01014\times CGDD}}$ | 0.994 | 2621.0 | 8.85 |
| | B4 | $Y = \dfrac{55460}{1+e^{11.72-0.01386\times CGDD}}$ | 0.976 | 2529.0 | 9.80 | $Y = \dfrac{60640}{1+e^{10.42-0.01326\times CGDD}}$ | 0.994 | 2444.0 | 8.80 |
| | MIB0 | $Y = \dfrac{26780}{1+e^{9.515-0.01242\times CGDD}}$ | 0.991 | 1193.0 | 11.49 | $Y = \dfrac{28090}{1+e^{11.44-0.01537\times CGDD}}$ | 0.999 | 521.2 | 4.66 |
| | MIB1 | $Y = \dfrac{52790}{1+e^{8.192-0.0112\times CGDD}}$ | 0.994 | 2152.0 | 7.55 | $Y = \dfrac{52330}{1+e^{10.63-0.01482\times CGDD}}$ | 0.997 | 1518.0 | 5.59 |
| | MIB2 | $Y = \dfrac{62180}{1+e^{8.256-0.01111\times CGDD}}$ | 0.984 | 4336.0 | 13.12 | $Y = \dfrac{56850}{1+e^{10.71-0.01505\times CGDD}}$ | 0.998 | 1503.0 | 5.03 |
| | MIB3 | $Y = \dfrac{77700}{1+e^{9.947-0.01246\times CGDD}}$ | 0.994 | 3199.0 | 8.07 | $Y = \dfrac{74910}{1+e^{11.25-0.01527\times CGDD}}$ | 0.988 | 4554.0 | 12.11 |
| | MIB4 | $Y = \dfrac{67700}{1+e^{12.19-0.01406\times CGDD}}$ | 0.996 | 2194.0 | 7.44 | $Y = \dfrac{66810}{1+e^{10.87-0.01453\times CGDD}}$ | 0.983 | 4780.0 | 14.48 |
| DM | F0 | $Y = \dfrac{2116}{1+e^{11.07-0.01148\times CGDD}}$ | 0.994 | 88.0 | 8.29 | $Y = \dfrac{2050}{1+e^{10.407-0.01349\times CGDD}}$ | 0.976 | 135.1 | 14.22 |
| | F1 | $Y = \dfrac{5457}{1+e^{9.379-0.01199\times CGDD}}$ | 0.983 | 392.9 | 12.81 | $Y = \dfrac{3667}{1+e^{9.273-0.01214\times CGDD}}$ | 0.979 | 226.6 | 12.92 |
| | F2 | $Y = \dfrac{3747}{1+e^{9.13-0.01116\times CGDD}}$ | 0.990 | 189.7 | 10.53 | $Y = \dfrac{5364}{1+e^{9.269-0.01303\times CGDD}}$ | 0.987 | 322.4 | 11.43 |
| | F3 | $Y = \dfrac{6464}{1+e^{13.34-0.01686\times CGDD}}$ | 0.988 | 387.3 | 11.52 | $Y = \dfrac{6939}{1+e^{10.04-0.01292\times CGDD}}$ | 0.984 | 464.6 | 14.33 |
| | F4 | $Y = \dfrac{5064}{1+e^{11.29-0.01314\times CGDD}}$ | 0.998 | 129.7 | 5.81 | $Y = \dfrac{4879}{1+e^{10.68-0.01326\times CGDD}}$ | 0.963 | 320.6 | 14.83 |
| | B0 | $Y = \dfrac{1885}{1+e^{9.652-0.01151\times CGDD}}$ | 0.987 | 112.0 | 12.11 | $Y = \dfrac{1909}{1+e^{8.651-0.01055\times CGDD}}$ | 0.957 | 123.1 | 14.85 |
| | B1 | $Y = \dfrac{4279}{1+e^{9.112-0.01132\times CGDD}}$ | 0.997 | 130.4 | 6.70 | $Y = \dfrac{2075}{1+e^{8.294-0.01061\times CGDD}}$ | 0.981 | 122.9 | 12.75 |
| | B2 | $Y = \dfrac{1942}{1+e^{9.059-0.01084\times CGDD}}$ | 0.965 | 133.0 | 14.74 | $Y = \dfrac{4364}{1+e^{9.691-0.01024\times CGDD}}$ | 0.993 | 192.2 | 9.72 |
| | B3 | $Y = \dfrac{5928}{1+e^{5.239-0.00632\times CGDD}}$ | 0.996 | 157.0 | 6.09 | $Y = \dfrac{4773}{1+e^{9.64-0.0102\times CGDD}}$ | 0.974 | 378.8 | 14.97 |
| | B4 | $Y = \dfrac{4576}{1+e^{10.37-0.01243\times CGDD}}$ | 0.996 | 157.6 | 7.43 | $Y = \dfrac{4419}{1+e^{10.26-0.01125\times CGDD}}$ | 0.979 | 327.8 | 14.72 |
| | MIB0 | $Y = \dfrac{6353}{1+e^{12.2-0.0137\times CGDD}}$ | 0.999 | 101.4 | 3.91 | $Y = \dfrac{3502}{1+e^{9.801-0.01233\times CGDD}}$ | 0.996 | 123.7 | 7.27 |

**Table 3.** *Cont.*

| Index | Treatment | 2022 | | | | 2023 | | | |
|---|---|---|---|---|---|---|---|---|---|
| | | Logistic Model | $R^2$ | RMSE | nRMSE/% | Logistic Model | $R^2$ | RMSE | nRMSE/% |
| | MIB1 | $Y = \dfrac{6106}{1+e^{6.47-0.00867\times CGDD}}$ | 0.984 | 358.5 | 11.58 | $Y = \dfrac{4940}{1+e^{9.148-0.01218\times CGDD}}$ | 0.990 | 254.6 | 11.44 |
| | MIB2 | $Y = \dfrac{7636}{1+e^{10.06-0.01321\times CGDD}}$ | 0.993 | 341.9 | 8.48 | $Y = \dfrac{6268}{1+e^{9.5-0.01083\times CGDD}}$ | 0.987 | 455.3 | 14.59 |
| | MIB3 | $Y = \dfrac{9939}{1+e^{6.338-0.00819\times CGDD}}$ | 0.994 | 407.3 | 8.40 | $Y = \dfrac{8858}{1+e^{10.82-0.01149\times CGDD}}$ | 0.984 | 618.7 | 13.69 |
| | MIB4 | $Y = \dfrac{3458}{1+e^{6.84-0.00829\times CGDD}}$ | 0.990 | 152.2 | 9.54 | $Y = \dfrac{7544}{1+e^{9.801-0.0114\times CGDD}}$ | 0.993 | 347.3 | 9.15 |

### 3.2. Dynamic Characteristics of Growth Index

The growth rate of pakchoi in four parts showed a unimodal curve with CGDD (Figures 4a–l and 5a–l). Under different organic fertilizer gradient treatments, the growth rate peak was higher than that under F0, B0, and MIB0 treatments, and the CGDD of the maximum growth rate was affected. When using fresh water irrigation, F2 increased the plant height maximum growth rate, F1 increased the leaf area index maximum growth rate, and F3 significantly increased the maximum growth rate of fresh weight and dry matter accumulation. When using brackish water irrigation, B3 increased the plant height maximum growth rate, B1 increased the leaf area index maximum growth rate, and both B3 and B4 increased the maximum growth rate of fresh weight and dry matter accumulation. When using magnetized–ionized brackish water irrigation, MIB3 increased the maximum growth rate of plant height, fresh weight, and dry matter accumulation, especially for fresh weight and dry matter accumulation maximum growth rate, and MIB1 increased the leaf area index maximum growth rate. The growth rate of fresh weight and dry matter accumulation was significantly higher under magnetized–ionized brackish water irrigation compared to brackish water irrigation. However, MIB4 suppressed the maximum growth rate of plant height and leaf area index.

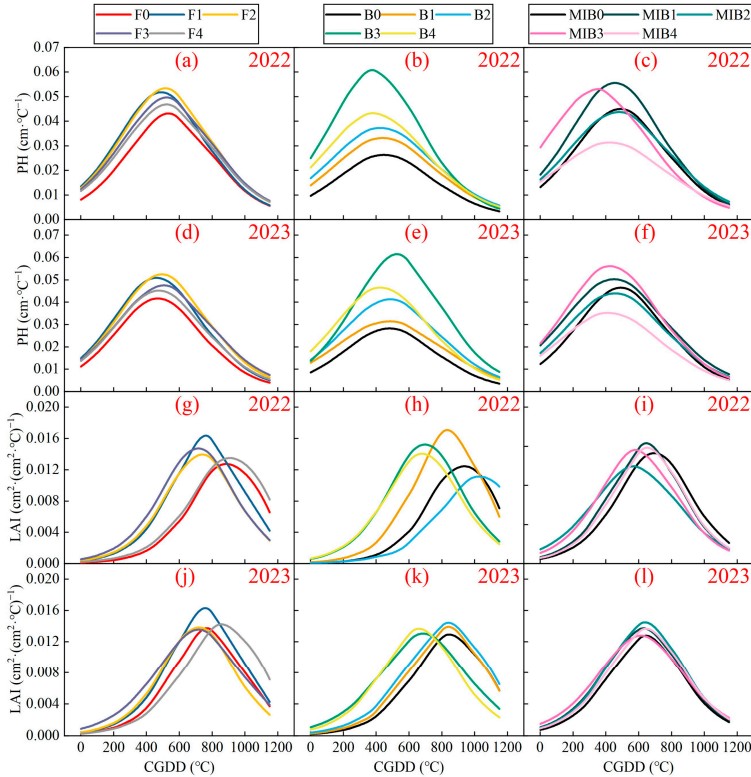

**Figure 4.** Plant height and leaf area index growth rates with cumulative growing degree days in 2022 and 2023.

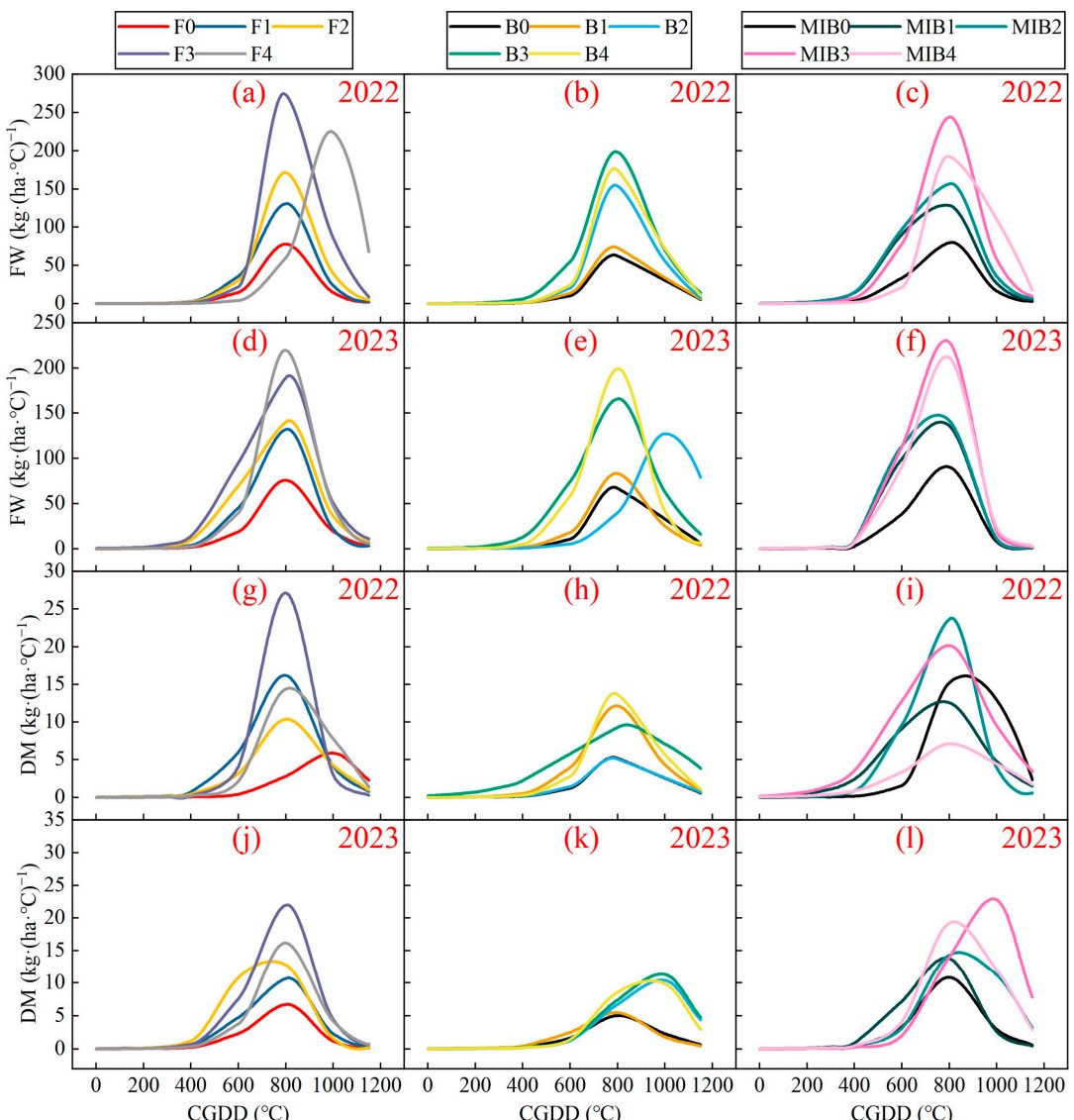

**Figure 5.** Fresh weight and dry matter accumulation growth rates with cumulative growing degree days in 2022 and 2023.

Figure 6a–h illustrates the trend of $CGDD_0$ with an increasing organic fertilizer gradient. The minimum $CGDD_0$ values for PH, LAI, FW, and DM were observed in the MIB3, MIB2, MIB2, and F2 treatments, respectively. In conclusion, the appropriate application of organic fertilizer can significantly increase the maximum growth rate of the pakchoi growth indices, reflecting the positive impact of organic fertilizer on pakchoi growth. However, it is crucial to avoid excessive application, as it may have adverse effects on pakchoi growth. Moreover, the choice of irrigation water type, especially MIB water, can further enhance the efficiency of organic fertilizer use and promote the fast growth period of pakchoi, as indicated by the shifting of the inflection point of the logistic curve.

The results from our analysis demonstrate that the characteristic parameters of the logistic model were influenced by both irrigation water types and organic fertilizer rates in both years, as shown in Tables A1 and A2. With the application of certain organic fertilizers, $CGDD_1$ decreases, $CGDD_2$ increases, and $\Delta CGDD$ increases, resulting in a longer duration of the fast growth period. Under fresh water irrigation, the minimum $CGDD_1$ treatments for PH, LAI, FW, and DM were F1, F3, F3, and F2, respectively. Under brackish water irrigation, the minimum $CGDD_1$ treatments for PH, LAI, FW, and DM were B4, B3, B3, and B1, respectively. Under magnetized–ionized brackish water irrigation, the

minimum $CGDD_1$ treatments for PH, LAI, FW, and DM were MIB3, MIB3, MIB1, and MIB1, respectively. These results suggest that the proper application of organic fertilizer can advance the onset of the fast growth period.

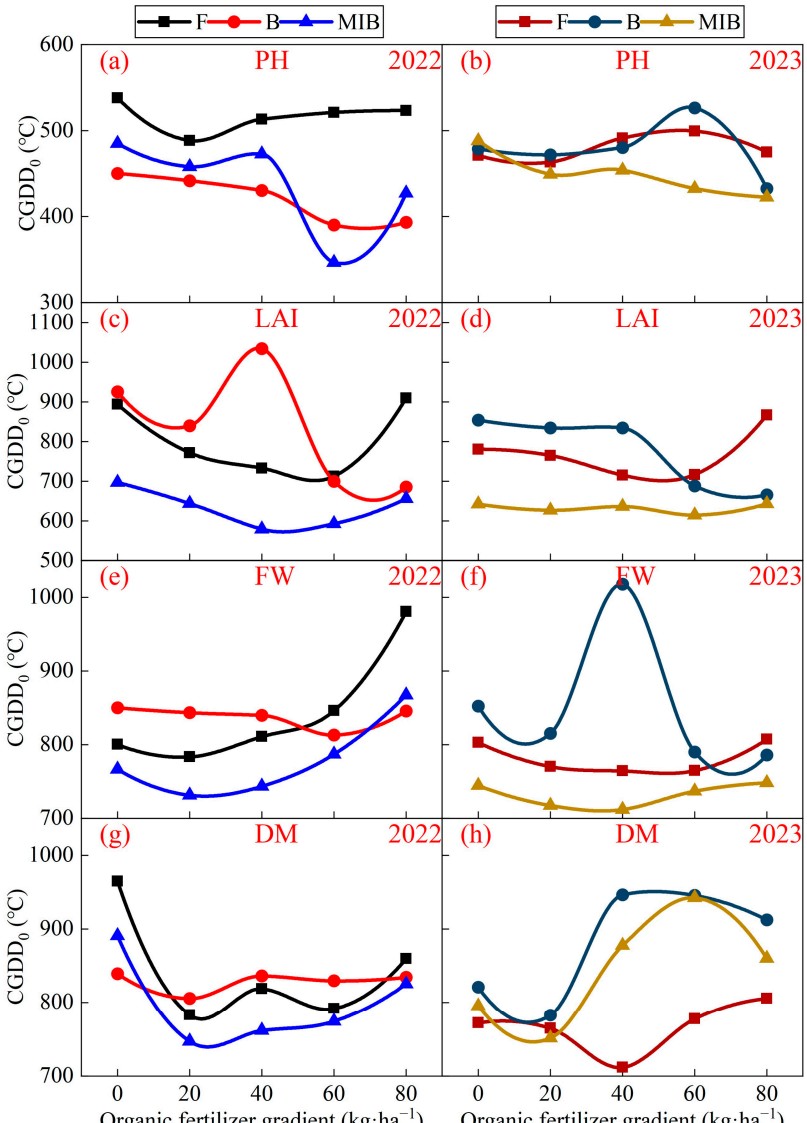

**Figure 6.** $CGDD_0$ with the organic fertilizer gradient.

Tables A1 and A2 demonstrate that in both years, the highest values of $\Delta CGDD$ were observed for PH, LAI, FW, and DM after using the MIB4, B3, B3, and B3 treatments, respectively, reaching 628.36 °C, 475.01 °C, 259.73 °C, and 416.82 °C, respectively. This suggests that while organic fertilizer application can promote early growth, a balanced approach is essential to avoid adverse impacts on plant development. The maximum values of $V_{max}$ for PH, LAI, FW, and DM were observed after using the B3, B1, F3, and F3 treatments, reaching 0.06 cm/°C, 0.017 $cm^2/cm^2 \cdot °C$, 313.03 kg/ha·°C, and 27.243 kg/ha·°C, respectively. The highest $V_{avg}$ values for PH, LAI, FW, and DM were observed after using the B3, B1, F3, and F3 treatments, reaching 0.53 cm/°C, 0.015 $cm^2/cm^2 \cdot °C$, 274.46 kg/ha·°C, and 23.886 kg/ha·°C, respectively. The maximum $Y_{Vmax}$ for PH, LAI, FW, and DM were observed after using the B3, F1, MIB3, and MIB3 treatments, respectively, reaching 23.29 cm, 4.81 $cm^2/cm^2$, 38,850 kg/ha, and 4970 kg/ha. These results suggest that $\Delta CGDD$, $V_{max}$, $V_{avg}$, and $Y_{Vmax}$ can be enhanced by applying 60 kg/ha of organic fertilizer, indicating

improved growth rates and yield potential for pakchoi. However, excessive fertilization can have an inhibitory effect.

### 3.3. Relationship between Yield Composition and Logistic Model Characteristics Parameters

$Y_{mmax}$, $Y_{cmax}$, $Y_{Vmax}$ corresponding to the PH (Figure 7a), FW (Figure 7c), and DM (Figure 7d) of pakchoi have a very strong positive correlation, with a correlation coefficient greater than 0.99. $V_{max}$ and $V_{avg}$ also have a strong positive correlation with $Y_{mmax}$, $Y_{cmax}$, and $Y_{Vmax}$, with a correlation coefficient of 0.88–0.94. In contrast, the correlation coefficient between $Y_{mmax}$, $Y_{cmax}$, and $Y_{Vmax}$ of LAI is 0.86, and the correlation coefficient between $V_{max}$, $V_{avg}$ and $Y_{mmax}$, $Y_{cmax}$, and $Y_{Vmax}$ is 0.58–0.66 (Figure 7b). The correlations between $CGDD_0$, $\Delta CGDD$, and other parameters are low or even negative. Yield is positively correlated with the $Y_{mmax}$, $Y_{cmax}$, $Y_{Vmax}$, $V_{max}$, and $V_{avg}$ of PH, FW, and DM and negatively correlated with $CGDD_0$ while being positively correlated with $\Delta CGDD$. Therefore, the growth index of pakchoi directly affects yield formation, and extending the duration of the fast growth period appropriately can increase yield.

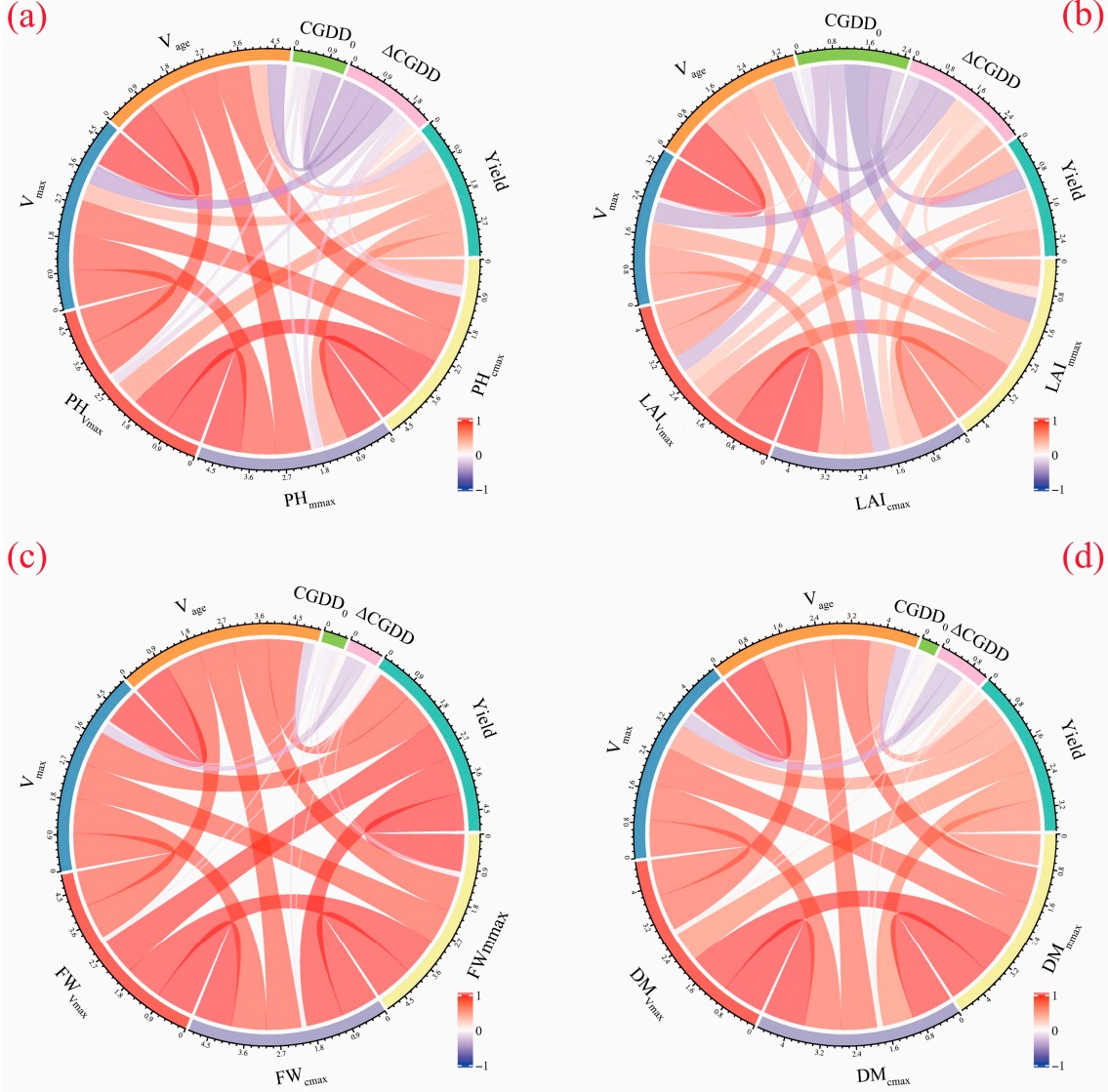

**Figure 7.** Correlation between yield and logistic model eigenvalues. (**a**) plant height model eigenvalues; (**b**) leaf area index model eigenvalues; (**c**) fresh weight model eigenvalues; (**d**) dry matter accumulation model eigenvalues.

Note: Yield, $Y_{mmax}$, $Y_{cmax}$, $Y_{Vmax}$, $V_{max}$, $V_{avg}$, $CGDD_0$, and $\Delta CGDD$ represent the fresh weight yield of pakchoi, measured maximum growth index, calculated maximum growth index, the corresponding value of the growth index at the maximum growth rate, maximum growth rate, average growth rate, cumulative growing degree days at the maximum growth rate, and duration cumulative growing degree days during the fast growth period of pakchoi, respectively.

While Figure 7 provides a general indication of the correlation between yield and various factors, it does not clearly illustrate the underlying mechanism of the impact of these factors on yield under different types of water irrigation. Hence, we selected growth index $Y_{mmax}$, logistic model characteristic parameter $V_{avg}$, and meteorological index $\Delta CGDD$ to construct structural equation models (SEM) for the plant height–leaf area index–yield level in fresh water (Figure 8), brackish water (Figure 9), and magnetized–ionized brackish water (Figure 10) with the addition of organic fertilizer based on the results mentioned above.

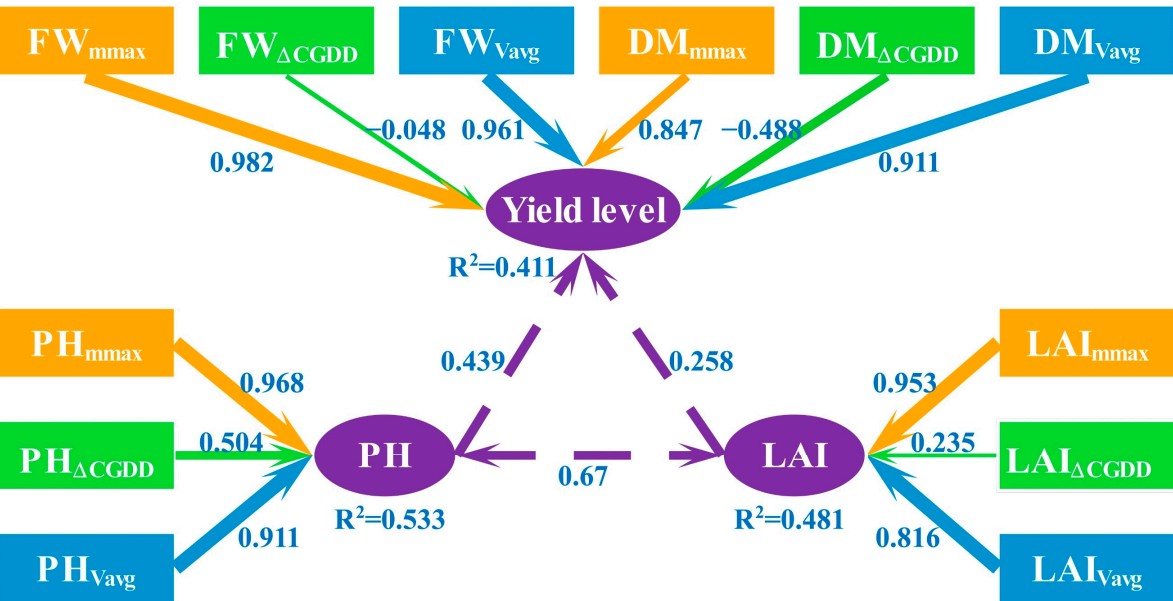

**Figure 8.** Plant height–leaf area index–yield level relationship model under fresh water irrigation.

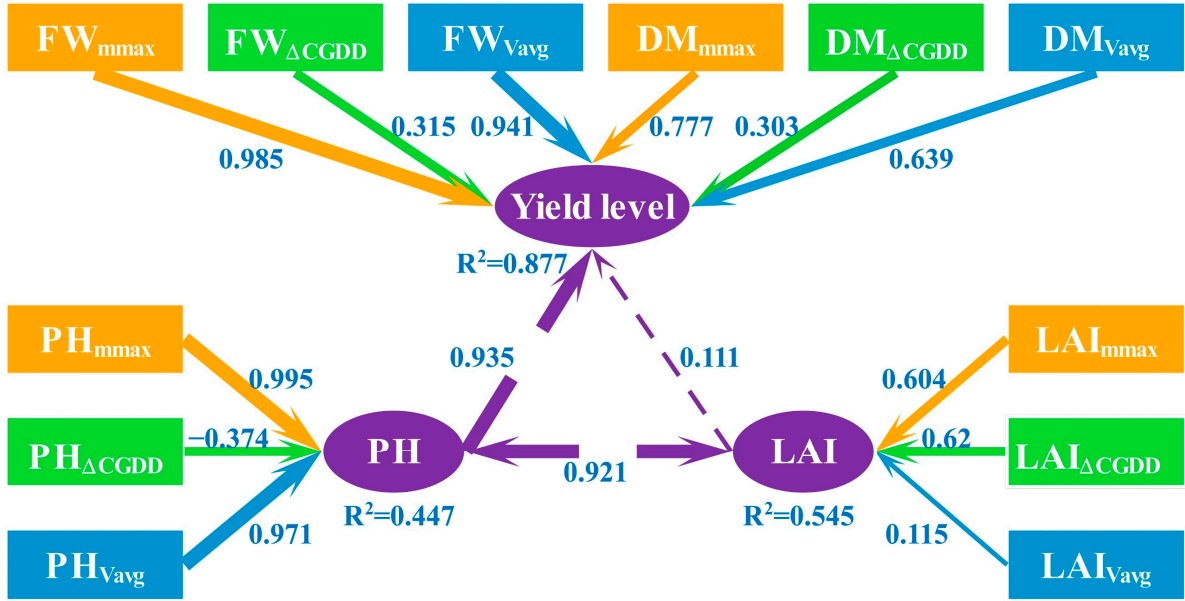

**Figure 9.** Plant height–leaf area index–yield level relationship model under brackish water irrigation.

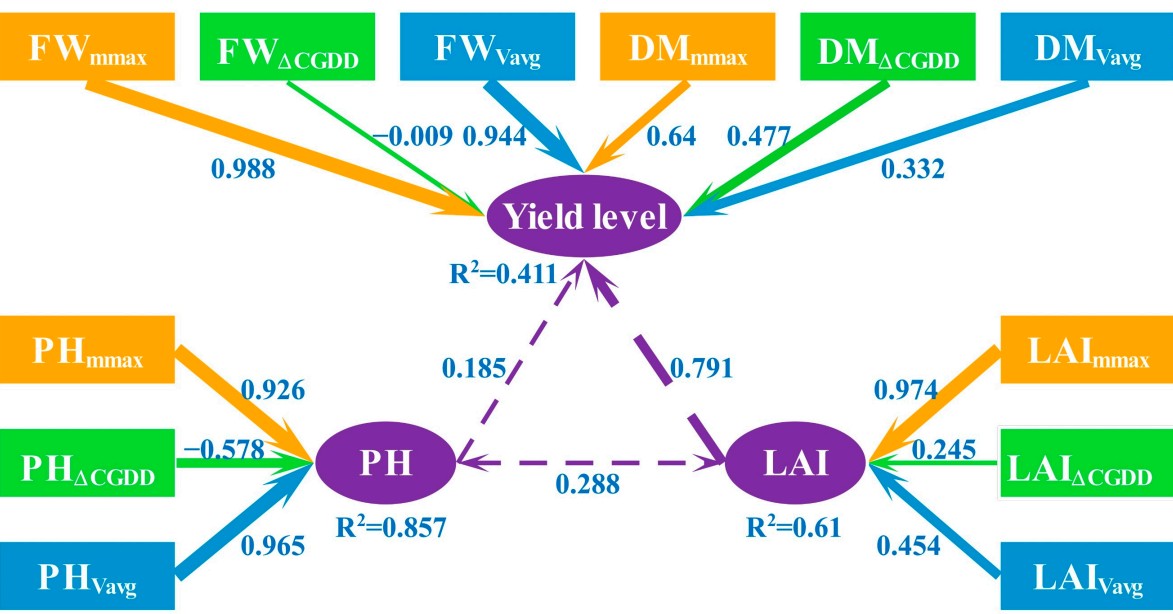

**Figure 10.** Plant height–leaf area index–yield level relationship model under magnetized–ionized brackish water irrigation.

The direct effects of PH and LAI on the yield level of pakchoi were positive, but their impact was limited. The effect of PH on yield level was greater than that of LAI under fresh water and brackish water irrigation, while the effect of LAI was greater than PH under magnetized–ionized brackish water irrigation. $PH_{mmax}$ and $LAI_{mmax}$ had the strongest direct effect on PH and LAI under fresh water irrigation, while $PH_{\Delta CGDD}$ and $LAI_{\Delta CGDD}$ had the smallest effect. $F_{mmax}$ had the largest direct effect on the yield level, while $DM_{mmax}$ had the smallest effect, and $FW_{\Delta CGDD}$ and $DM_{\Delta CGDD}$ had negative effects (Figure 8). Under brackish water irrigation, $PH_{mmax}$ and $LAI_{\Delta CGDD}$ had the greatest direct effect on PH and LAI, while $PH_{Vavg}$ and $LAI_{Vavg}$ had the smallest effect, and $PH_{\Delta CGDD}$ had a negative direct effect on PH. $F_{mmax}$ had the largest direct effect on the yield level, while $DM_{\Delta CGDD}$ had the smallest effect (Figure 9). Under magnetized–ionized brackish water irrigation, $PH_{Vavg}$ and $LAI_{mmax}$ had the strongest direct effect on PH and LAI, while $PH_{mmax}$ and $LAI_{\Delta CGDD}$ had the smallest effect, and $PH_{\Delta CGDD}$ had a negative direct effect on PH. $F_{mmax}$ had the largest direct effect on the yield level, while $DM_{Vavg}$ had the smallest effect, and $FW_{\Delta CGDD}$ had a negative effect (Figure 10).

From our investigation of the different irrigation scenarios, it is clear that $Y_{mmax}$ has a more considerable direct impact on structural variables compared to meteorological variable $\Delta CGDD$, which has a smaller or even negative impact. The specific effects of organic fertilizer on pakchoi growth characteristics differ under various irrigation conditions. However, the direct influence of $Y_{mmax}$ on structural variables remains a consistent trend.

In conclusion, significant differences were observed in the effects of organic fertilizer on the growth characteristics of pakchoi under different types of water irrigation. However, overall, the observed variable $Y_{mmax}$ had a greater direct impact on the structural variables, whereas $\Delta CGDD$ had a smaller or even negative impact.

## 4. Discussion

### 4.1. Effects of Combined Application of Organic Fertilizer on Growth Indexes and Dynamic Characteristics

The long-term application of organic fertilizer has been proven to improve soil fertility by increasing the content of available nutrients, including alkaline-hydrolyzable nitrogen, available phosphorus, and available potassium. The short-term application of organic fertilizer also has a positive impact on soil fertility, particularly in nutrient-deficient soils [14,50]. This is because it can provide available nutrients for plants, improve soil structure, and

enhance water retention ability, all of which can affect the growth and yield of pakchoi. Numerous studies have demonstrated that the application of organic fertilizer can significantly improve the growth rate and yield of pakchoi while reducing leaf fall and plant mortality during growth [41,51]. Additionally, organic fertilizer use has been shown to increase the number and diversity of microorganisms in soil, improve soil permeability, and enhance water retention, thus enhancing photosynthesis and nutrient absorption, resulting in increased PH and LAI. In this study, we found that the different application rates of organic fertilizer improved the growth index characteristics of pakchoi. Our results showed that the effects of organic fertilizer combined with brackish water irrigation on PH, LAI, and growth rate were more pronounced compared to fresh water and magnetized–ionized brackish water irrigation. Specifically, our data revealed a significant difference in maximum PH between the B0, B1, B2, B3, and B4 treatments in 2022 and 2023 (Figure 2b,e). Moreover, variations in the timing of the maximum growth rate of the LAI were observed among different treatments (Figure 4h). In summary, the combined application of organic fertilizer with brackish water irrigation had a more pronounced effect on PH and LAI.

Organic fertilizer is a rich source of essential nutrients (such as nitrogen, phosphorus, and potassium) that facilitate plant growth and development [52]. The utilization of organic fertilizer has been demonstrated to enhance nitrogen utilization efficiency, crop yield, and crop quality [53]. Its application in vegetable cultivation has gained widespread recognition among scholars. Numerous researchers have reported that applying organic fertilizer not only leads to an increase in the plant height and individual weight of pakchoi but also results in elevated levels of essential components such as vitamin C, soluble sugars, and soluble proteins in pakchoi. Conversely, the application of organic fertilizer has been shown to reduce nitrate content [54–56]. The combined application of organic fertilizer with fresh water, brackish water, and magnetized–ionized brackish water irrigation had a substantial effect on the fresh weight, dry weight, and growth rate of pakchoi. These findings align with those of previous researchers [51,57]. Specifically, when the amount of organic fertilizer applied was 60 kg/ha, the improvement effect was notably higher than that of other treatments. Nonetheless, since organic fertilizer is a slow-release fertilizer, it cannot supply adequate nutrients during the early growth stage of pakchoi. As a result, the growth rate of fresh weight and dry weight in the initial growth stage was limited and maintained a steady trend, as demonstrated in Figure 5.

In conclusion, these results emphasize the importance of carefully selecting organic fertilizer rates and irrigation water types to optimize the growth dynamics of pakchoi. Properly managed organic fertilizer application can advance the onset of the fast growth period, enhancing growth rates and yield potential. However, excessive fertilization should be avoided to prevent inhibitory effects on crop development. These findings contribute to our understanding of how agronomic practices can be tailored to improve crop growth and yield in pakchoi cultivation.

*4.2. The Relationship between Characteristic Parameters of Logistic Model and Structural Equation Model*

The logistic equation is characterized by a typical S-shaped curve composed of piecewise exponential, linear, and convex functions that vary with independent variables, and its eigenvalues are primarily used to describe the linear accumulation period [58,59]. However, its simulation accuracy may be reduced, particularly during periods of fast growth. In this study, the logistic model proved to be highly accurate in simulating the growth process, which not only highlights the importance of crop growth models in describing crop growth processes but also directly reflects the growth dynamics of pakchoi. Since growth rates are a direct reflection of carbon assimilation capacity and the distribution of photosynthetic products, they can be largely manipulated via agronomic practices [60,61]. Consequently, the correlation between growth rates and yield-related parameters varies. In this study, the $V_{max}$ and $V_{avg}$ of FW exhibited the highest correlation with yield, whereas the $V_{max}$ and $V_{avg}$ of LAI showed the lowest correlation with yield. Moreover, a significant positive

correlation was observed between the eigenvalues of the growth index model and pakchoi yield, whereas the correlation between the eigenvalues of CGDD and yield was relatively weak (Figure 7).

Zhang et al. conducted a comprehensive investigation that involved using path analysis to elucidate the varying weights of cotton growth rates across different time periods and plant parts. Initially, a logistic model was employed to examine the influence of planting density and nitrogen application on biomass formation. However, this model provided limited insights into the relationship between planting density and cotton yield. To address this gap, the researchers extended their analysis by integrating the logistic model with path analysis. This combined approach allowed for a quantitative and intuitive representation of how the biomass formation process impacts cotton yield [62]. In this paper, the structural equation models revealed the direct effects of PH and LAI on yield levels under different irrigation conditions. Although Figure 7 provides an initial depiction of the relationship between influencing factors and yield formation, it does not provide specific information on the extent to which they affect yield. Therefore, a structural equation model was established to quantify and demonstrate the effect of growth index formation on yield level for different types of irrigation water combined with organic fertilizer. As shown in Figure 7, there is a positive correlation between yield and the change in $\Delta$CGDD, but the structural equation model reveals that changes in $FW_{\Delta CGDD}$ and $DM_{\Delta CGDD}$ negatively impact the yield level under freshwater irrigation. Additionally, the direct effect of $PH_{\Delta CGDD}$ on PH under brackish water and magnetized–ionized brackish water irrigation was negative. Furthermore, $FW_{\Delta CGDD}$ had a negative effect on the yield level under magnetized–ionized brackish water irrigation. In conclusion, these findings emphasize the importance of considering the type of irrigation when optimizing the growth and yield of pakchoi. The structural variables of plant height and leaf area index, along with yield-related factors, play crucial roles. Additionally, the duration of the fast growth period, as represented by $\Delta$CGDD, should be carefully managed to achieve the desired yield outcomes.

## 5. Conclusions

In this study, it was observed that the application of 60 kg/ha of organic fertilizer positively influenced the growth of pakchoi, as indicated by improvements in various biological parameters. However, excessive fertilization at a rate of 80 kg/ha resulted in the inhibition of the plant height, leaf area index, and maximum growth rate of pakchoi. Magnetized–ionized brackish water irrigation proved to be a more favorable approach for enhancing pakchoi growth, resulting in significantly higher rates of fresh weight and dry matter accumulation compared to conventional brackish water irrigation. The direct effects of different influencing factors on yield levels varied across different types of irrigation water. Nevertheless, it was evident that the maximum values of growth indexes exerted a significant direct influence on plant height, leaf area index, and yield levels. Conversely, the duration of cumulative growing degree days during the fast growth period of pakchoi had a relatively minor effect or potentially negative impact. Based on these findings, the appropriate application of additional organic fertilizer holds significant importance in enhancing pakchoi yield.

**Author Contributions:** S.L. (Shudong Lin): Conceptualization, Formal analysis, Investigation, Resources, Writing—original draft; C.W.: Data curation, Formal analysis; Q.L.: Resources, Methodology; K.W.: Formal analysis, Resources, Data curation; Q.W.: Conceptualization, Methodology, Visualization, Supervision, Writing—review and editing; M.D.: Methodology, Supervision; L.S.: Methodology, Software; S.L. (Shiyao Liu): Formal analysis, Software; X.D.: Resources, Data curation. All authors have read and agreed to the published version of the manuscript.

**Funding:** This study was jointly supported by the Major Science and Technology Projects of the Autonomous Region (2020A01003–3).

**Data Availability Statement:** The data presented in this study are available upon request from the corresponding author.

**Conflicts of Interest:** The authors declare that they have no known competing financial interests or personal relationships that could have appeared to influence the work reported in this paper.

## Appendix A

**Table A1.** Characteristics and parameters of the logistic model for the plant height and leaf area index dynamic change process in 2022 and 2023.

| Index | Treatment | 2022 | | | | | | | 2023 | | | | | | |
|---|---|---|---|---|---|---|---|---|---|---|---|---|---|---|---|
| | | $CGDD_0$ | $CGDD_1$ | $CGDD_2$ | $\Delta CGDD$ | $V_{max}$ | $V_{avg}$ | $Y_{Vmax}$ | $CGDD_0$ | $CGDD_1$ | $CGDD_2$ | $\Delta CGDD$ | $V_{max}$ | $V_{avg}$ | $Y_{Vmax}$ |
| PH | F0 | 537.92 | 296.31 | 779.54 | 483.23 | 0.042 | 0.037 | 15.43 | 471.14 | 228.29 | 713.99 | 485.69 | 0.042 | 0.037 | 15.38 |
| | F1 | 488.26 | 239.70 | 736.82 | 497.12 | 0.051 | 0.045 | 19.41 | 463.28 | 214.96 | 711.60 | 496.64 | 0.051 | 0.045 | 19.22 |
| | F2 | 513.02 | 259.88 | 766.16 | 506.28 | 0.053 | 0.046 | 20.20 | 491.09 | 236.96 | 745.21 | 508.24 | 0.052 | 0.046 | 20.09 |
| | F3 | 520.87 | 258.95 | 782.79 | 523.84 | 0.049 | 0.043 | 19.34 | 499.24 | 227.33 | 771.14 | 543.82 | 0.047 | 0.041 | 19.38 |
| | F4 | 523.24 | 261.13 | 785.34 | 524.21 | 0.046 | 0.040 | 18.26 | 474.85 | 215.69 | 734.01 | 518.32 | 0.045 | 0.039 | 17.71 |
| | B0 | 450.09 | 178.19 | 721.99 | 543.80 | 0.026 | 0.023 | 10.87 | 478.51 | 217.02 | 739.99 | 522.97 | 0.028 | 0.025 | 11.18 |
| | B1 | 441.46 | 151.06 | 731.85 | 580.80 | 0.033 | 0.029 | 14.62 | 471.55 | 165.18 | 777.92 | 612.74 | 0.031 | 0.027 | 14.53 |
| | B2 | 430.22 | 133.27 | 727.18 | 593.91 | 0.037 | 0.033 | 16.83 | 480.03 | 200.06 | 760.00 | 559.94 | 0.041 | 0.036 | 17.45 |
| | B3 | 389.93 | 136.09 | 643.77 | 507.68 | 0.060 | 0.053 | 23.29 | 526.27 | 272.49 | 780.04 | 507.56 | 0.060 | 0.053 | 23.14 |
| | B4 | 393.15 | 104.06 | 682.24 | 578.18 | 0.043 | 0.038 | 18.93 | 432.41 | 162.78 | 702.05 | 539.27 | 0.047 | 0.041 | 19.08 |
| | MIB0 | 484.72 | 224.06 | 745.38 | 521.32 | 0.045 | 0.039 | 17.71 | 488.10 | 237.59 | 738.61 | 501.02 | 0.046 | 0.040 | 17.57 |
| | MIB1 | 457.69 | 197.77 | 717.62 | 519.85 | 0.056 | 0.049 | 21.96 | 449.14 | 157.90 | 740.37 | 582.47 | 0.050 | 0.044 | 22.23 |
| | MIB2 | 472.60 | 183.65 | 761.54 | 577.89 | 0.044 | 0.038 | 19.09 | 453.79 | 167.84 | 739.73 | 571.88 | 0.044 | 0.039 | 19.08 |
| | MIB3 | 346.45 | 63.41 | 629.49 | 566.08 | 0.053 | 0.046 | 22.65 | 432.32 | 163.63 | 701.00 | 537.37 | 0.056 | 0.049 | 22.94 |
| | MIB4 | 427.03 | 112.85 | 741.22 | 628.36 | 0.031 | 0.027 | 14.92 | 422.49 | 127.79 | 717.20 | 589.41 | 0.035 | 0.031 | 15.76 |
| LAI | F0 | 893.82 | 702.51 | 1085.13 | 382.62 | 0.013 | 0.012 | 3.81 | 779.97 | 586.78 | 973.16 | 386.38 | 0.013 | 0.012 | 3.95 |
| | F1 | 770.97 | 576.90 | 965.03 | 388.12 | 0.016 | 0.014 | 4.68 | 764.41 | 565.38 | 963.44 | 398.06 | 0.016 | 0.014 | 4.81 |
| | F2 | 732.68 | 538.96 | 926.40 | 387.44 | 0.014 | 0.012 | 4.12 | 714.53 | 520.55 | 908.51 | 387.96 | 0.014 | 0.012 | 4.12 |
| | F3 | 712.13 | 511.57 | 912.70 | 401.13 | 0.015 | 0.013 | 4.51 | 716.29 | 484.23 | 948.35 | 464.12 | 0.013 | 0.012 | 4.70 |
| | F4 | 909.48 | 698.80 | 1120.17 | 421.37 | 0.014 | 0.012 | 4.40 | 866.68 | 654.85 | 1078.52 | 423.68 | 0.014 | 0.012 | 4.57 |
| | B0 | 924.61 | 742.08 | 1107.15 | 365.07 | 0.013 | 0.011 | 3.57 | 853.56 | 650.95 | 1056.18 | 405.23 | 0.013 | 0.011 | 3.96 |
| | B1 | 839.58 | 656.00 | 1023.16 | 367.16 | 0.017 | 0.015 | 4.76 | 833.98 | 626.43 | 1041.52 | 415.08 | 0.014 | 0.012 | 4.28 |
| | B2 | 1033.54 | 821.26 | 1245.82 | 424.56 | 0.011 | 0.010 | 3.61 | 833.71 | 611.45 | 1055.97 | 444.52 | 0.014 | 0.012 | 4.78 |
| | B3 | 698.99 | 499.65 | 898.32 | 398.67 | 0.015 | 0.013 | 4.65 | 687.83 | 450.32 | 925.33 | 475.01 | 0.013 | 0.011 | 4.62 |
| | B4 | 685.05 | 482.50 | 887.61 | 405.11 | 0.014 | 0.012 | 4.33 | 665.31 | 456.65 | 873.98 | 417.33 | 0.013 | 0.012 | 4.26 |
| | MIB0 | 696.77 | 495.45 | 898.09 | 402.64 | 0.014 | 0.012 | 4.34 | 642.26 | 439.08 | 845.44 | 406.36 | 0.013 | 0.011 | 3.87 |
| | MIB1 | 643.48 | 446.08 | 840.88 | 394.80 | 0.015 | 0.013 | 4.50 | 626.57 | 413.85 | 839.30 | 425.46 | 0.014 | 0.012 | 4.39 |
| | MIB2 | 578.80 | 342.73 | 814.86 | 472.13 | 0.012 | 0.011 | 4.42 | 636.23 | 427.23 | 845.23 | 418.00 | 0.014 | 0.013 | 4.54 |
| | MIB3 | 592.80 | 382.37 | 803.23 | 420.86 | 0.014 | 0.013 | 4.61 | 614.06 | 381.48 | 846.64 | 465.17 | 0.013 | 0.011 | 4.51 |
| | MIB4 | 655.90 | 459.63 | 852.17 | 392.54 | 0.015 | 0.013 | 4.35 | 643.01 | 431.99 | 854.02 | 422.03 | 0.013 | 0.012 | 4.31 |

**Table A2.** Characteristics and parameters of the logistic model for the fresh weight and dry matter accumulation dynamic change process in 2022 and 2023.

| Index | Treatment | 2022 | | | | | | | 2023 | | | | | | |
|---|---|---|---|---|---|---|---|---|---|---|---|---|---|---|---|
| | | $CGDD_0$ | $CGDD_1$ | $CGDD_2$ | $\Delta CGDD$ | $V_{max}$ | $V_{avg}$ | $Y_{Vmax}$ | $CGDD_0$ | $CGDD_1$ | $CGDD_2$ | $\Delta CGDD$ | $V_{max}$ | $V_{avg}$ | $Y_{Vmax}$ |
| FW | F0 | 800.11 | 711.11 | 889.10 | 177.99 | 77.431 | 67.891 | 10,465 | 802.66 | 702.48 | 902.85 | 200.37 | 75.649 | 66.329 | 11,510 |
| | F1 | 783.15 | 688.61 | 877.68 | 189.07 | 132.136 | 115.856 | 18,970 | 769.91 | 673.25 | 866.57 | 193.31 | 137.476 | 120.538 | 20,180 |
| | F2 | 810.70 | 719.68 | 901.72 | 182.04 | 172.217 | 150.999 | 23,805 | 763.91 | 646.13 | 881.68 | 235.55 | 145.757 | 127.799 | 26,070 |
| | F3 | 845.67 | 765.09 | 926.26 | 161.17 | 313.031 | 274.464 | 38,310 | 764.56 | 644.93 | 884.18 | 239.25 | 195.960 | 171.817 | 35,600 |
| | F4 | 980.21 | 888.14 | 1072.29 | 184.15 | 228.419 | 200.276 | 31,940 | 806.96 | 716.36 | 897.56 | 181.20 | 220.002 | 192.897 | 30,270 |
| | B0 | 849.91 | 748.13 | 951.69 | 203.56 | 68.771 | 60.298 | 10,630 | 851.93 | 751.30 | 952.55 | 201.25 | 74.765 | 65.554 | 11,425 |
| | B1 | 843.25 | 738.78 | 947.72 | 208.94 | 78.504 | 68.832 | 12,455 | 815.08 | 714.57 | 915.59 | 201.02 | 83.859 | 73.527 | 12,800 |
| | B2 | 839.65 | 747.27 | 932.03 | 184.76 | 166.403 | 145.901 | 23,345 | 1017.31 | 896.72 | 1137.90 | 241.18 | 127.940 | 112.177 | 23,430 |
| | B3 | 812.85 | 701.93 | 923.77 | 221.84 | 199.288 | 174.735 | 33,570 | 789.57 | 659.70 | 919.43 | 259.73 | 166.262 | 145.777 | 32,790 |
| | B4 | 845.48 | 750.47 | 940.48 | 190.01 | 192.197 | 168.517 | 27,730 | 785.76 | 686.45 | 885.07 | 198.62 | 201.037 | 176.268 | 30,320 |
| | MIB0 | 766.16 | 660.12 | 872.21 | 212.09 | 83.145 | 72.901 | 13,390 | 744.16 | 658.49 | 829.83 | 171.33 | 107.957 | 94.656 | 14,045 |
| | MIB1 | 731.23 | 613.68 | 848.79 | 235.11 | 147.852 | 129.635 | 26,395 | 717.13 | 628.28 | 805.97 | 177.69 | 193.922 | 170.091 | 26,165 |
| | MIB2 | 743.32 | 624.74 | 861.89 | 237.14 | 172.658 | 151.386 | 31,090 | 711.77 | 624.25 | 799.29 | 175.05 | 213.855 | 187.507 | 28,425 |
| | MIB3 | 786.70 | 682.54 | 890.85 | 208.31 | 245.610 | 215.349 | 38,850 | 736.74 | 650.49 | 822.98 | 172.49 | 285.969 | 250.736 | 37,455 |
| | MIB4 | 867.00 | 773.33 | 960.67 | 187.33 | 237.966 | 208.647 | 33,850 | 748.11 | 657.47 | 838.74 | 181.27 | 242.687 | 212.787 | 33,405 |
| DM | F0 | 964.37 | 849.64 | 1079.10 | 229.46 | 6.072 | 5.324 | 1058 | 771.35 | 673.74 | 868.96 | 195.22 | 6.915 | 6.063 | 1025 |
| | F1 | 782.50 | 672.62 | 892.37 | 219.75 | 16.352 | 14.337 | 2729 | 763.96 | 655.47 | 872.46 | 217.00 | 11.128 | 9.757 | 1834 |
| | F2 | 818.25 | 700.22 | 936.28 | 236.06 | 10.452 | 9.164 | 1874 | 711.52 | 610.43 | 812.62 | 202.19 | 17.469 | 15.317 | 2682 |
| | F3 | 791.32 | 713.20 | 869.44 | 156.24 | 27.243 | 23.886 | 3232 | 776.91 | 675.00 | 878.82 | 203.82 | 22.418 | 19.656 | 3470 |
| | F4 | 859.14 | 758.93 | 959.36 | 200.43 | 16.637 | 14.587 | 2532 | 805.25 | 705.95 | 904.54 | 198.59 | 16.178 | 14.184 | 2440 |
| | B0 | 838.79 | 724.35 | 953.24 | 228.90 | 5.423 | 4.755 | 943 | 820.31 | 695.43 | 945.19 | 249.75 | 5.033 | 4.413 | 955 |
| | B1 | 805.02 | 688.67 | 921.37 | 232.70 | 12.109 | 10.617 | 2140 | 782.08 | 657.90 | 906.27 | 248.37 | 5.501 | 4.824 | 1038 |
| | B2 | 835.70 | 714.21 | 957.19 | 242.98 | 5.263 | 4.614 | 971 | 946.02 | 817.46 | 1074.58 | 257.12 | 11.176 | 9.799 | 2182 |
| | B3 | 829.09 | 620.67 | 1037.50 | 416.82 | 9.365 | 8.211 | 2964 | 945.19 | 816.06 | 1074.32 | 258.25 | 12.170 | 10.671 | 2387 |
| | B4 | 834.00 | 728.09 | 939.92 | 211.83 | 14.224 | 12.472 | 2288 | 912.00 | 794.94 | 1029.06 | 234.13 | 12.428 | 10.897 | 2210 |
| | MIB0 | 890.32 | 794.21 | 986.42 | 192.21 | 21.764 | 19.082 | 3177 | 794.89 | 688.08 | 901.70 | 213.62 | 10.795 | 9.465 | 1751 |
| | MIB1 | 746.42 | 594.49 | 898.36 | 303.87 | 13.232 | 11.601 | 3053 | 751.01 | 642.89 | 859.12 | 216.23 | 15.044 | 13.190 | 2470 |
| | MIB2 | 761.31 | 661.65 | 860.98 | 199.33 | 25.226 | 22.118 | 3818 | 877.36 | 755.73 | 998.98 | 243.25 | 16.967 | 14.877 | 3134 |
| | MIB3 | 773.59 | 612.85 | 934.33 | 321.48 | 20.358 | 17.849 | 4970 | 941.93 | 827.29 | 1056.58 | 229.30 | 25.438 | 22.304 | 4429 |
| | MIB4 | 824.99 | 666.15 | 983.83 | 317.68 | 7.168 | 6.284 | 1729 | 859.74 | 744.21 | 975.26 | 231.05 | 21.500 | 18.851 | 3772 |

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
