# Peer review of "Effects of Combined Application of Organic Fertilizer on the Growth and Yield of Pakchoi under Different Irrigation Water Types"

_agronomy, doi:10.3390/agronomy13102468_

Round 1

Reviewer 1 Report

The research idea, the title of the paper, and the objectives of the research have great potential, but the written paper itself is based more on statistics and statistical models which put the results and the applicability and the objective of the research. In fact, the quality of the research and thus the paper is reflected not only in well-executed statistics but also in clear interpretation, which I consider the main shortcoming of this paper. Also, I did not understand whether you considered fertilization and the three types of irrigation as a monofactorial or a two-factorial experiment.

Also, it would be advisable to include at least the basic elements of mineral composition (N, P, K, the nitrates you mentioned) in addition to the morphological characteristics, since this is fertilization with organic manure, and it would also be useful to include the basic chemical composition (e.g., antioxidant capacity, perhaps glucosinolates, etc.).

My personal impression is that you tried to "hide" the lack of data (mineral composition and chemistry) with strong statistics, and that there was no connection to the experiment itself and to agronomy in general (especially because you did not link the results and the discussion).

Reviewer 2 Report

Mineral fertilization is an inherent element of modern agriculture. It allows you to obtain high yields and may also have a beneficial effect on their quality. Of course, the one-sided use of large doses of mineral fertilizers, especially nitrogen fertilizers, and the abandonment of organic fertilization have negative effects. Therefore, it is important to select appropriate doses based on the nutritional needs of plants and to use organic fertilization. An important problem related to crop production, especially in some regions, is the need for irrigation of crops and the resulting soil salinization. Therefore, I believe that the research presented in the manuscript is interesting and up-to-date. Thematically, the manuscript falls within the thematic scope of the journal ‘Agronomy’. The experiments presented in the manuscript were properly planned, it allows to explain the hypotheses put forward in the introduction. The research material is sufficient. The results were statistically analyzed and discussed and interpreted in detail. Tables and graphs are generally well prepared, understandable and legible.

I just have a few comments to consider

1. line [41-42] – is ‘…and low fertilizer utilization rates can significantly impact agricultural output, leading to food crises and jeopardizing human survival and development’  This is not entirely true. In many countries/regions of the world, excessive use of mineral fertilizers is a greater threat to the environment or food quality. The problem of soil salinity is similar. It is not important everywhere. Authors should indicate which regions this applies to

2. Line [120] – is ‘….901 m..’ – what it mean? 900 m above sea level?

3. Table 1 – Soil pH was not determined?

4. line [141] - where do such doses of organic fertilization come from? On what basis were these quantities determined?

5. line [147] - what fertilizer was used, what and how many ingredients were introduced into the soil?

6. line [156-159] – when measurements began and how often and at what intervals they were taken.

7. Line [176] - In my opinion, with such a small plot area (3 m2), converting the yield to 1 hectare is not correct. It is better to give the yield per 1 m2.

8. Line [204] - I suggest putting the units in brackets, this will increase the readability of the text – ‘…..represents the plant height (cm), leaf area index (cm2/cm2),…’

9. line [462-463] - such a conclusion does not result from the research conducted. The authors used mineral fertilization. The results show that the use of additional organic fertilization is beneficial.

10. I suggest considering moving tables 4 and 5 to supplementary materials

Reviewer 3 Report

The manuscript titled “Effects of Combined Application of Organic Fertilizer on the Growth and Yield of Pakchoi under Different Irrigation Water Types” submitted by Quanjiu Wang et al. describing the influence of organic fertilizer on the growth index of pakchoi, and monitor the application of organic fertilizer in the production of pakchoi under different irrigation water types. This is a well-written article and I anticipate that the manuscript should be of great interest to the researchers working on organic fertilizers and irrigation water. I considered the manuscript suitable for publication subject to following improvements.

The abstract section should be improved by indicating the implications of the findings. Similarly elaborate the statement (Line 20-22)“In this period, the duration cumulative growing degree days (ΔCGDD) reached 20 maximum values of 628.36 (MIB4), 475.01 (B3), 259.73 (B3), and 416.82 (B3) for plant 21 height, leaf area index, fresh weight, and dry matter accumulation, respectively” to make it readers friendly.

Line 26: You can use abbreviation instead of full form (MIB water) to avoid repetition.

Line 31-32: Have you used or compared your findings with any inorganic fertilizer? If yes, then mention it here.

In Introduction section, add relevant references in Line 51-52 “The traditional methods of irrigation lead to water waste and soil salinization, whereas traditional fertilizer use decreases land fertility and causes the accumulation of chemical residues, resulting in environmental pollution”.

In Introduction section, add some latest references from the available literature.

Results section is well written and presented.

The discussion is short and descriptive required more information and comparison with other reports.

Round 2

Reviewer 1 Report

Dear Sir/Madam, I have read Your reply to the reviewers and can see that You have taken all my comments into account and corrected them in the paper. In my opinion, the article is now suitable for publication in this journal.